

# High-resolution digital mapping of soil organic carbon in permafrost terrain using machine-learning: A case study in a sub-Arctic peatland environment

Matthias B. Siewert[12]

5 [1]Department of Physical Geography, Stockholm University, Stockholm, SE-106 91, Sweden
[2]New Affiliation: Department of Ecology and Environmental Science, Umeå University, Umeå, SE-901 87, Sweden

*Correspondence to*: Matthias B. Siewert (matthias.siewert@umu.se)

**Abstract.** Soil organic carbon (SOC) stored in northern peatlands and permafrost affected soils are key components in the global carbon cycle. I quantify SOC stocks in a sub-arctic mountainous peatland environment in the discontinuous permafrost zone in Abisko, northern Sweden. Four machine-learning techniques are evaluated: multiple linear regression, artificial neural networks, support vector machine and random forest. The random forest approach performed best and was used to predict SOC for several depth increments at a spatial resolution of $2 \times 2$ m. A high-resolution ($1 \times 1$m) land cover classification generated for this study is the most relevant predictive variable. The landscape mean SOC storage (0–150 cm) is estimated to $7.9 \pm 8.0$ kg C m$^{-2}$ and the SOC stored in the top meter (0–100 cm) to $7.0 \pm 6.3$ kg C m$^{-2}$. The predictive modeling highlights the relative importance of wetland areas and in particular peat plateaus for the landscape SOC storage. A surprising large number of small scale wetland areas are mapped forming very local hot-spots of SOC storage. The results show that robust SOC predictions are possible with the available methods and very high-resolution remote sensing data. Strong environmental gradients associated with land cover and permafrost distribution are the most challenging methodological aspect. However, in this study, at local, regional and circum-Arctic scale the main factor limiting robust, high-resolution SOC mapping efforts is the scarcity of soil pedon data from across the entire environmental space. For the Absiko region, past SOC and permafrost dynamics indicate that most of the SOC is barely 2000 years old and very dynamic in wetland areas with permafrost related landforms. Future research needs to investigate the geomorphic response of permafrost degradation and the fate of SOC across all landscape compartments in post-permafrost landscapes.





## 1 Introduction

High-latitudes are among the regions most affected by increasing temperatures and climate change (IPCC, 2013). Large amounts of soil organic carbon (SOC) and the abundance of wetlands as a substantial source of methane ($CH_4$), are factors that make these regions a key component in the global carbon (C) cycle (McGuire et al., 2009). Frozen conditions, cold

temperatures and water-logging are characteristics of wetlands, peatlands and permafrost affected soils that reduce decomposition rates of SOC (Davidson and Janssens, 2006; Ping et al., 2015). This has led to the accumulation of large stocks of SOC in high-latitude ecosystems (Tarnocai et al., 2009). SOC stocks in the circumpolar permafrost region are currently estimated to ~1300 Pg (Hugelius et al., 2014) and represent around half of the global SOC stocks (Köchy et al., 2015). High-resolution mapping efforts are necessary to map SOC in these environments due to the great spatial and vertical

variability of permafrost-affected soils (Siewert et al., 2015, 2016). A significant proportion of this SOC is stored in northern wetland and peatland areas (Gorham, 1991). However, warming temperatures, environmental changes caused by warming of soils and consequent permafrost degradation are projected to lead to a gradual and prolonged release of greenhouse gases in the future (Schuur et al., 2015).

This article investigates SOC storage and longterm SOC dynamics in the Abisko region, sub-Arctic Sweden, where

numerous ecosystem dynamics related to climate warming have been documented (Callaghan et al., 2013). Dramatic changes in peat mires have been reported between 1970 and 2000 (Malmer et al., 2005). During this period, degradation of permafrost and vegetation changes can be associated with increases in landscape scale $CH_4$ emissions (Christensen et al., 2004). An analysis of present day C fluxes indicates that losses from soil and over the hydrosphere currently offset C accumulation in peatlands and above ground biomass (Lundin et al., 2016). To improve our understanding of permafrost C

dynamics, particularly over longer timescales, high-resolution maps of landscape distribution and partitioning of SOC, including the vertical partitioning, and an integration into numerical models is necessary (Mishra et al., 2013; Schuur et al., 2015). Combined with a better temporal framework of past C dynamics, this will improve the projection of future global temperatures.

Thematic maps are a commonly used to upscale SOC from point measurements to landscape scale in permafrost

environments (Hugelius, 2012). This method has been used in combination with soil maps to estimate SOC storage in the circumpolar permafrost region (Hugelius et al., 2014; Tarnocai et al., 2009). Land cover maps have been used at local to regional scales to estimate SOC values in numerous circumpolar environments (Fuchs et al., 2015; Hugelius et al., 2010, 2011; Hugelius and Kuhry, 2009; Palmtag et al., 2015; Siewert et al., 2015; Zubrzycki et al., 2013). While soil maps may better reflect soil properties and soil forming processes, a land cover classification (LCC) has the advantage that it can be

readily generated from remote sensing data using the spatial resolution of the respective sensor. However, thematic mapping also represents a strong generalization, as equal soil properties are assumed for all areas covered by the same mapping class. Furthermore, for land cover maps there is an implicit assumption that land cover alone reflects below-ground soil properties (Hugelius, 2012).

Predictive modeling of SOC values can yield well resolved pixel based estimates and provide a potential improvement over

thematic mapping. Quantitative methods in soil science are largely based on the works of Jenny (1941, 1980) and have significantly developed since. Digital soil mapping is the successor of these concepts using modern methods. A comprehensive summary has been published by McBratney et al. (2003) and many examples are available for example in





Boettinger et al. (2010). However, at higher-latitudes the adoption of these methods has been slow and only few studies apply predictive modeling methods to upscale soil properties in sub-Arctic and Arctic permafrost environments (Bartsch et al., 2016; Baughman et al., 2015; Ding et al., 2016; Mishra and Riley, 2012, 2014; Pastick et al., 2014). This has several reasons including: the limited availability of environmental input data, the limited amount of soil pedon data (Mishra et al.,

2013) and the large local scale variability of permafrost-affected soils (Siewert et al., 2016). To cope with these limitations new mapping methods for permafrost environments are necessary to better constrain SOC stocks in the northern circumpolar permafrost region (Mishra et al., 2013). Such new methods include the use of machine-learning in digital soil mapping (Hastie et al., 2009; Li et al., 2011; McBratney et al., 2003).

This study aims to compare a variety of machine-learning techniques for the prediction of SOC in permafrost and peatland

environments and to estimate carbon storage under different land cover types. Combined with radiocarbon dates to estimate SOC accumulation, this will advance our knowledge on SOC distribution and longterm C dynamics in high-latitude permafrost environments. Machine-learning in soil science covers a set of data-mining techniques that can recognize patterns in large data-sets and learn from these to predict quantitative soil variables. Many algorithms are available and robust prediction results are possible (Hastie et al., 2009; Li and Heap, 2008; Li et al., 2011). The workflow of this study is

outlined in Fig. 1. A 1 × 1 m very high-resolution LCC is generated. Four prediction models are compared: a multiple linear regression (MLR) model, an artificial neural network (ANN), a support vector machine (SVM) and the random forests (RF). The best performing model is used to demonstrate a high-resolution (2 × 2 m) spatial regression modeling approach of SOC pedon data to landscape scale. The LCC is used for stratified extraction of the SOC per class. As an outcome, I provide high-resolution SOC storage data for a key sub-Arctic research site on ecosystem adaption to climate change in Abisko,

northern Sweden. This includes insights on the spatial and vertical partitioning of the SOC and its association with different environmental variables. The temporal evolution of the SOC stocks over the Holocene is interpreted from eight radio-carbon dates and the future development of SOC stocks in high-latitude environments is discussed.

Figure 1 near here

**2 Study area**

The study area is a sub-Arctic mountain environment in the Abisko region near Stordalen along the shores of lake Torneträsk in northernmost Sweden (Fig. 2 and Fig. 3). Environmental monitoring and research has been conducted for more than a century in the region and a particular interest has been the main peatland complex called Stordalen mire (Callaghan et al., 2013; Jonasson et al., 2012). The mapping extent covers two major peatland complexes, Stordalen and

Storflaket, east of the Abisko Scientific Research Station, the surrounding birch forest and adjacent alpine tundra zone. The altitude ranges from 342 m a.s.l. corresponding to the lake level of Torneträsk to 932 m a.s.l. in the mountain zone. The total mapping area is 65 km².

A mean annual air temperature of 0.5°C (2002–2011) and a mean annual precipitation of 332 mm have been measured for the period 2002–2011 in Abisko. These values are approximate indicators as topography and rain-shadow effects have

strong local influence in the region (Callaghan et al., 2013).

Wetland soils in the study area are of organic nature (Histosols). Soils in the surrounding forest have mostly characteristics of Podzols or micro-Podzols with a bleached horizon below the organic surface layer. The alpine soils are often limited to





shallow surface organic layers over rock (Leptosols) or very weakly developed soils in unconsolidated slope or moraine material (Regosols). However, if permafrost occurs within 1 m at higher elevation or when cryoturbation occurs and permafrost occurs within 2 m, then soils classify as Cryosols (FAO, 2015).

The study area is located in the zone of discontinuous permafrost (Brown et al., 1997). The onset of late Holocene permafrost aggradation in Stordalen was around 2650 cal BP with a first phase that lasted until 2100 cal BP and a second phase after ca. 700 cal BP (Kokfelt et al., 2010). Today, the occurrence of permafrost at lower elevations is confined to peat mires due to insulation effects of the peat. Here it can be several meters thick below elevated permafrost peat plateaus (palsa) (Johansson et al., 2011). At higher elevation, permafrost was modeled to occur above 850 m a.s.l. on north-east and east-facing slopes and above 1000 to 1100 m a.s.l. on west and south facing slopes west of Abisko Station (Ridefelt et al.,

2008). However, widespread permafrost degradation has occurred at least since the 1980s. This was associated with increased active layer thicknesses and complete disappearance of permafrost in some parts of Stordalen mire (Åkerman and Johansson, 2008) and with a decrease in permafrost thickness from 15 m in 1980 to 9 m 2009 in one borehole (Åkerman and Johansson, 2008; Johansson et al., 2011).

Figure 2 and Figure 3 near here

### 3 Methods

#### 3.1 Field survey and SOC data

Soil sampling was performed in September 2013 and June 2015. In total, 47 sites were sampled following initial field reconnaissance. Sampling was undertaken along 4 main transects with 8–10 points of equal distance. These transects were

laid out as a semi-random sampling schemes to represent major environmental gradients with a restricted amount of sampling points in difficult terrain. These were complemented with smaller transects (n= 2 & 3) and six additional profiles from land covers with small patch sizes otherwise not covered. Some transects are incomplete due to points located in lakes, and to avoid disturbance of experimental installations or wildlife.

The sampling procedure followed (Schoeneberger et al., 2012), with an additional protocol for permafrost-affected soils

(Ping et al., 2013; Siewert et al., 2016) Sampling in peat was performed in 5 cm intervals by cutting samples of known volume from the open pit, using fixed volume cylinders or in case of waterlogged conditions using a fixed volume Russian peat corer. The permafrost was sampled by hammering a steel pipe into the frozen ground. Sampling in the surrounding birch forest and tundra was performed according to soil horizons. The organic layer (OL) was sampled completely, while deeper soil horizons were sampled in 5 to 10 cm intervals. Soils outside the peat complexes are in general very shallow and

have large volumes of coarse fragments and often become impossible to sample after ~20–50 cm as the fractured lithic contact is reached. The unsampled coarse fraction consisting of unconsolidated bedrock was noted and used to correct the amount of soil material. For the tundra heath, the OL can be discontinuous with patches of vegetation alternating with patches of bare ground. Here the SOC storage was corrected for the proportional coverage of the OL per m². Each point location was recorded in the field using a hand-held GPS device (±5 m location accuracy).

A total of 278 individual soil samples were collected. Dry bulk density (DBD, g cm$^{-3}$) was calculated from oven dried soil samples at 65°C for 5 days. The loss on ignition method was performed on all samples at 550°C for 5 hrs to determine the organic matter (OM) content and at 950°C for 2 hrs to determine the inorganic C content (Heiri et al., 2001). C% was



measured for a subset of 73 samples using an EA 1110 Elemental Analyzer (CE Instruments, Italy). A further subset of samples with high inorganic C were acid treated but showed very little reaction and LOI 950°C indicated very low inorganic C content in the soils of 0.73 ± 0.62% for all samples and hence was not further analyzed. C% values were then used to predict C% for samples were only LOI was available using a third order polynomial regression model (Fuchs et al., 2015;
Hugelius et al., 2011; Siewert et al., 2015). The SOC storage was calculated per soil sample using C%, DBD, excluding soil material of the coarse fraction (<2 mm, CF, %) and sample depth interval. Depth intervals that have not been sampled were gap filled based on soil horizon information. The average pedon depth was 103 ± 29 cm for wetland pedons and 26 ± 27 cm for non-wetland pedons. To estimate the total SOC (SOC–Tot) stored in the landscape, all wetland pedons were processed to a reference depth of 1.5 m and non-wetland pedons to a depth of 1 m. If the pedon did not reach that depth it was
extrapolated based on a trend in the pedon or similar pedons or set to zero if the lithic contact was reached. This interpolation is necessary to provide standardized and consistent input data for the predictive modeling approach. Other depth intervals that were extracted are the stored in the organic surface layer (SOC–OL), the SOC stored for the top 30 cm (SOC 0–30) and the SOC for the top 100 cm (SOC 0–100). The depth of the organic surface layer (OL–Depth) was also predicted.
To evalutate longterm SOC dynamics, eight samples were submitted for AMS $^{14}$C dating to the Radiocarbon Laboratory in Poznan, Poland. The resulting dates were calibrated to calendar years, cal yr BP (1950) using OxCal 4.2 (Bronk Ramsey, 2016).

### 3.2 Environmental datasets

A set of spatially referenced environmental datasets is used to reflect ecosystem properties in the study area. In the optical domain, an orthophoto of 1 m spatial resolution with RGB-bands from 2008 (© Lantmäteriet, I2014/00691) and a SPOT5 orthorectified multispectral satellite image (Path 045, row 208, acquired 10.08.2013)(© Lantmäteriet, I2014/00691) were available. The spectral bands of the SPOT image include green, red and near-infrared (NIR) at 10 m spatial resolution and a shortwave-infrared (SWIR) band at 20 m spatial resolution. A topographical correction for differential illumination was
applied to the orthophoto and the SPOT image to compensate for terrain shadows using a "Minnaert correction with slope" implementation (Law and Nichol, 2004). The illumination corrected SPOT image was used to derive the normalized difference vegetation index (NDVI) (Rouse et al., 1974) and the soil-adjusted vegetation index (SAVI) with an L-value of 0.7 (Huete, 1988). Furthermore, the ratio of NIR/SWIR bands is used. A digital elevation model (DEM) of 2 m spatial resolution (© Lantmäteriet, I2014/00691) was used to generate several derivative topographic datasets. These include slope,
aspect, profile and plan curvature, topographic ruggedness index (TRI), topographic position index (TPI) (Wilson et al., 2007), a TPI based landform classification (Guisan et al., 1999), and topographic wetness index (TWI) (Moore et al., 1991). Survey based vector maps with a scale of 1:250 000 were obtained for the geology and quaternary land cover (© SGU, I2014/00691) and for vegetation (© Lantmäteriet, I2014/00691). Geospatial analyses as well as raster and vector processing was performed using GDAL/OGR (GDAL, 2016), SAGA (Conrad et al., 2015), Orfeo toolbox and R (R Core Team, 2017)
softwares.





### 3.3 Land cover classification

An object-based approach was used to generate a detailed LCC (Blaschke, 2010). The LCC is used as a predictor variable and for stratified extraction of the digital soil mapping results. First, the orthophoto was combined with the DEM at 1 m spatial resolution. A segmentation layer was generated by grouping pixels into homogeneous areas with a minimum region

size of 130 m². From this a water mask was classified in a separate step using the red band of the orthophoto and a slope layer. A land cover training set was created by combining field survey information with visual interpretation of the orthophoto and topography. The following layers were used as input for the classification algorithm: the orthophoto, elevation and slope; the SPOT5 4-band satellite image and NDVI (Rouse et al., 1974), SAVI (Huete, 1988) and NIR/SWIR (SPOT5 derivatives). The ratio of NIR/SWIR band can be beneficial to separate bedrock and bare rock areas (Andersson,

2016). The segments were then classified using a support vector machine (SVM; Chang and Lin, 2011) algorithm. Artificial surfaces were hand digitized and masked out. The latter mainly includes a road and railway passing through the study area. The individual thematic classes are adapted from Andersson (2016) and described in Table A.1 in the supplement.

The quality of the classification was assessed using a set of 108 ground control points. These include the locations of the soil sampling sites and points along pathways collected in equal distance from the starting point. The kappa coefficient and

the overall accuracy are calculated for all land covers excluding water and artificial areas (Congalton, 1991).

### 3.4 Digital soil mapping using machine-learning

This article investigates the general applicability of machine-learning in the specific context of regression techniques for the mapping of SOC in high-latitude permafrost and peatland environments. Numerous machine-learning algorithms and

approaches exist. A comprehensive general overview on machine-learning techniques is provided by Hastie et al. (2009) and the general use of different machine-learning algorithms for digital soil mapping is thoroughly discussed for instance in McBratney et al. (2003), Li et al. (2011), Were et al. (2015) and Taghizadeh-Mehrjardi et al. (2016). Thus, only a brief description follows. Four commonly used machine-learning techniques were compared: a multiple linear regression (MLR) model, an artificial neural network (ANN) (Ripley, 1996), a support vector machine (SVM) (Chang and Lin, 2011) and

random forest (RF) (Breiman, 2001).

**Multiple linear regression (MLR)** assumes that the regression function defining the soil variable is linear or can be approximated using a linear equation. In a linear regression model the soil variable *f(x)* represents the dependent variable and the environmental predictors the independent variables $X_i$. Where $a$ is the intercept and $b_i$ are regression coefficients.

$$f(X) = a + \sum_{i=0}^{n} b_i x_i \tag{1}$$

The training data is used to define the regression equation and then used to predict the soil variable for unseen occurrences in the environmental space. MLR is a popular technique that is comparatively simple. In situations with limited input data and low-signal to-noise ratio, linear regression models can sometimes outperform non-linear methods (Forkuor et al., 2017; Hastie et al., 2009).





For the MLR model the *lm* function in R was used (R Core Team, 2017). A 10-fold cross-validation with five repetitions was trained to develop stable models of SOC–Tot using the 'caret' R package (Forkuor et al., 2017; Kuhn, 2008).

**Artificial neural network (ANN)** is a technique that simulates the biological nervous system. For continuous soil variables, it is a two-stage regression model typically represented by a network diagram with three layers. A layer of input cells represent environmental covariates and transmits it to a layer of hidden cells, which forwards it again to an output layer representing the soil property to be predicted. The units between the layers are connected by synapses. These connections, also called weights, define the model. The model simulates learning from examples by training the network iteratively with information about the conditions in which a certain value of the soil variable occurs. During each iteration the connection between the input layer, hidden layer and the output unit is adjusted. Finally, the trained model is used to predict soil properties of unvisited pixels (Behrens et al., 2005; Hastie et al., 2009).

The ANN was parameterized using a grid search approach for the variables defining the *size* of the hidden layer and the *decay* of weights in the neural network. The tuning was started with a value of 2 for *size* to avoid a local minimum. The 'caret' R package was used in combination with a 10-fold cross-validation with five repetitions to develop a stable model of SOC–Tot based on the smallest RMSE value (Forkuor et al., 2017; Kuhn, 2008).

**Support vector machine (SVM)** is a technique that generates an optimal separating hyperplane to differentiate classes that overlap and are not separable in a linear way. In this case, a large, transformed feature space is created to map the data with the help of kernel functions to separate it along a linear boundary. While initially developed for classification purposes, this technique can also be used for regression problems (Hastie et al., 2009; Vapnik, 1998).

For SVM a ε-regression with a gaussian radial basis kernel was used. This kernel can be considered a good general purpose kernel (Zeileis et al., 2004). The cost parameter $C$ and the $\varepsilon$ error threshold parameter were determined using the grid-search method. This was combined with a 10-fold cross-validation with five repetitions implemented by the 'caret' package in R (Forkuor et al., 2017; Kuhn, 2008).

**Random Forest (RF)** is a learner that combines decision tree and bagging methods (Breiman, 2001). RF draws a number of bootstrap samples ($n_{tree}$) from the input dataset representing individual soil samples and grows a large amount of unpruned regression trees (e.g. 500), where at each node random samples ($m_{try}$) of the environmental predictors are chosen. It then averages the prediction of all trees to predict new data (Liaw and Wiener, 2002).

For RF the randomForest R package was used (Liaw and Wiener, 2002). The default values for $m_{try}$ equaling the 3/predictors, a node size of 5 and $n_{tree}$ = 500 provided stable and visual sound results. Different variations of the parameters $m_{try}$, $n_{tree}$, node size and maximum node number were tested, but generate inferior visual results, while providing only minor improvements to $R^2$, but increasing the dependency of the most important predictive variable indicating overfitting. It is known that RF does not need extensive fine-tuning which can lead to overfitting (Ließ et al., 2016), thus the standard settings were applied. Sampling was performed with replacement and bias correction was applied to decrease overestimation for low values and underestimation for high values. To achieve stable model results, a 10-fold cross-validation with five repetitions was applied (Forkuor et al., 2017; Kuhn, 2008).

**Model selection and validation**

First, the performance of all four machine-learning algorithms (MLR, ANN, SVM and RF) to predict SOC–Tot was evaluated. For the validation, each model was trained using a 80% random split of the soil pedon dataset. The models are





then assessed using an internal validation of predicted values against the training dataset and an external validation using the remaining 20 % split as an unseen control dataset. The models are compared based on three commonly used error criteria for the internal and external validation. The error criteria include the coefficient of determination ($R^2$), the root mean squared error (RMSE) and Lin's concordance correlation coefficient (CCC) (Lin, 1989). Maps were developed by applying

predictive models of the entire soil pedon dataset to the environmental datasets. These were then visually examined for further model evaluation and compared to a thematic map of the SOC storage based on the combination of the LCC and average values per LCC class.

Originally, all models overestimated SOC contents for bare ground surfaces, such as lowland blockfields and stone beaches along the lake shore. To address this overestimation, 10 pseudo-training samples with 0.0 kg C m$^{-2}$ SOC were added at bare

ground locations identified in the orthophoto. These were distributed across the study area and kept to a low number to avoid strong bias of the training dataset. A similar approach has been used by Siewert et al. (2012) to support spatial interpolation of limited line measurements to estimate sediment thickness of talus cones. The performance of the models is assessed with and without these 10 pseudo-training samples. For the final analysis, the best performing algorithm (RF) was chosen to model and develop maps for each of the following soil variables depths: SOC–OL, SOC 0–30, SOC 0–100, SOC–

Tot and the OL–Depth. For this the entire soil pedon dataset and all environmental predictors were used. Predicted SOC values below 0 were set 0.





## 4 Results

### 4.1 Land cover classification

The LCC showed good agreement with the classes that have been observed in the field and areas that have been visually identified in the orthophoto (Fig. 4a and b). The accuracy assessment against ground control points collect in the field
results in a Kappa value of 0.71 and an overall accuracy of 74% (Table A.2). This does not include any water or artificial surfaces. These values are comparable to other high-latitude LCC accuracy assessments (Schneider et al., 2009; Siewert et al., 2015; Virtanen et al., 2004; Virtanen and Ek, 2014).

The object-based classification has a minimum patch size of 130 m², which was found to best differentiate areas of homogeneous land cover, while preserving characteristic shapes of landforms  relevant for SOC storage in high-latitudes.
This avoids miss-classification of individual pixels that can be problematic with pixel-based classification approaches at very high-resolution and reduces the need for post-processing filtering and other generalization methods (Siewert et al., 2015). Reese et al. (2014, 2015) demonstrated for the Abisko area, that very detailed pixel based vegetation maps are possible by combining laser scanning point clouds and SPOT5 satellite imagery. However, the chosen patch-size seemed more realistic to reflect most-likely homogeneous soil properties at this scale, both as input to predict SOC content and for
the stratified extraction of the predicted SOC values as applied here.

Figure 4 near here

### 4.2 Performance of four machine-learning algorithms to predict SOC

Four different models were compared to predict SOC–Tot stocks. Table 1 presents the results of the internal and external validation. There are large discrepancies between the models. RF consistently achieves the highest coefficient of determination ($R^2$) and CCC, while having the lowest root mean squared error (RMSE). This was followed by SVM with slightly lower performance for each error criteria. MLR and ANN achieved reasonable results for the internal validation. Only RF showed tantamount performance for the internal and external validation ($R^2 = 0.939$ compared to 0.908). SVM
showed acceptable results, while MLR and ANN fail to match values of the internal validation. The exclusion of the pseudo sampling points influences in particular the external validation for which the results show a significant drop in performance.

Table1 near here

All models underestimate large values and overestimate low values of SOC-Tot (Fig. 5). This so called regression to mean effect is a known shortcoming of the RF algorithm and was addressed using bias correction option in the randomForest package (Liaw and Wiener, 2002; Zhang and Lu, 2012). Yet, a slight overestimation for low values from ~0–25 kg C m$^{-2}$ and an underestimation for SOC–Tot values above ~60 kg C m$^{-2}$ remains.  Furthermore, RF cannot forecast values that are beyond the training dataset. Thus, even if the environmental variables suggest higher SOC stocks, there is no trend
extrapolation. In this study, the highest SOC value was 90 kg C m$^{-2}$ and cannot be exceeded in the model. This likely underestimates SOC values for some areas with thicker peat deposits than the 138 cm measured in our transect sampling, as





the thickness of peat can be up to 3 m in the mire (Malmer and Wallén, 1996). However, this also prevents gross overestimation of SOC.

Figure 5 near here

The developed output maps show significant differences for the four prediction models (Fig. 4 d-g). Major wetland areas can be recognized in all four maps. Wetlands and peat bogs generally represent areas with significantly higher SOC storage compared to surrounding soils (e.g. Hugelius et al., 2011; Siewert et al., 2015). The MLR model (Fig. 4d) shows very strong gradients from non-wetland areas in SOC storage. These seem to be underestimated opposed to wetland areas that are predicted to have very high SOC values throughout, compared to an expected distribution of SOC following a thematic map

(Fig. 4c). The ANN model (Fig. 4e) dos not reflect well the field situation of strong contrasts between wetland and non-wetland areas. Birch forest areas and blockfields seem in general highly overestimated with SOC–Tot values of ~20-30 kg C m$^{-2}$ compared to $4.2 \pm 2.2$ kg C m$^{-2}$ for the average of soil pedons in birch forest. Wetland areas show low values compared to the other models. SVM and RF visually correspond best to the field situation of sampled and analyzed soils as exemplified in the thematic map of SOC (Fig. 4d). SVM seems to slightly overestimate low SOC values close for lowland

birch forest and bare ground areas (Fig. 4f). RF is the only prediction model that can replicate the extent of elevated peat bogs and plateaus (Fig. 3 and Fig. 4g). RF manages to represent the contrast of bare ground areas and blockfields with SOC values typically <1 kg C m$^{-2}$.

**4.3  Environmental controls on SOC distribution**

As the RF model provided the best performance, the remaining analysis proceeds using only RF. The importance of each input variable for the prediction of the SOC–Tot is presented in Fig. 6. The LCC is the most important predictive variable. The strong dependence on the land cover classification likely reflects sensitivity to land cover segmentation. This is followed by a group of SPOT 5 input variables, elevation, TWI and slope. The SPOT5 variables include the NIR band, the

ratio of NIR/SWIR, NDVI and SAVI. These variables are related to the sensitivity of the included bands to vegetation, but also to bare ground cover signature (Andersson, 2016; Huete, 1988; Rouse et al., 1974). Elevation likely reflects the lapse rate of the mean annual air temperature gradient in steep mountainous terrain. The contribution of the TWI data is likely to identify waterlogged conditions along streams and in mires. This is likely supported by slope and is particularly evident for alpine willow communities that are located along flow accumulation pathways. Other variables seem to have equally little

predictive power. Exclusion of these predictive variables was tested, but generate an overestimation for the lower range of SOC values.

The variable importance changes for the individual prediction models of SOC 0–30, SOC 0–100, SOC–OL, Depth–OL (Fig. A.1) and SOC–Tot. However, the pattern usually resembles that of SOC–Tot. Land cover is the most important environmental variable in all models except for SOC 0–30, where NIR/SWIR is the most important variable with the LCC,

SAVI, NIR, and NDVI following in one group.

Figure 6 near here





### 4.4 SOC stocks landscape partitioning and age

Table 2 shows the landscape partitioning of the sampled SOC pedon values and the predicted values. Landscape mean SOC–Tot storage is predicted to $7.9 \pm 8.0$ kg C m$^{-2}$ and to $7.0 \pm 6.3$ kg C m$^{-2}$ for the top meter (0–100 cm; SOC 0–100) of soil. This compares to $5.8 \pm 0.5$ kg C m$^{-2}$ for the SOC–Tot using the land cover class for thematic mapping and

$5.3 \pm 0.5$ kg C m$^{-2}$ for the interval 0–100 cm. The highest SOC stock per class is estimated for the Sphagnum covered wetlands areas ($38.5 \pm 9.3$ kg C m$^{-2}$) followed by the other wetland classes: peat bog ($35.3 \pm 9.4$ kg C m$^{-2}$), lowland shrub wetland ($36.4 \pm 7.8$ kg C m$^{-2}$), sedge wetland ($33.6 \pm 9.9$ kg C m$^{-2}$) and forested wetland ($28.6 \pm 7.2$ kg C m$^{-2}$). The alpine willow class stores the highest amount of SOC of the remaining non-wetland classes with $9.2 \pm 3.2$ kg C m$^{-2}$, followed by birch forest ($7.9 \pm 4.3$ kg C m$^{-2}$) and dwarf-shrubs ($7.5 \pm 4.0$ kg C m$^{-2}$). The bare ground class stores the lowest amount of

SOC with $1.1 \pm 2.2$ kg C m$^{-2}$. This represents most likely an overestimation and should be close to <0.1 kg C m$^{-2}$. For SOC–OL, SOC 0–30 and SOC 0–100 similar patterns emerge. Permafrost was encountered in six pedons of which 4 were located in the peat bog with an average depth of $50 \pm 20$ cm, one in the sphagnum wetlands and one alpine tundra heath. The mire permafrost soils were sampled in early September in 2013, while the alpine heath tundra (AL–Depth = 37 cm) was sampled in June and does not represent maximum annual active layer depth. The partition of SOC stored in permafrost is

$10.1 \pm 13.3$ kg C m$^{-2}$ for peat bog soil samples and $8.5 \pm 12.1$ kg C m$^{-2}$ for Sphagnum wetland samples. This equals $0.2 \pm 0.0$ kg C m$^{-2}$ of the total landscape SOC weighted by area using thematic mapping. Prediction of the SOC in permafrost using digital soil mapping techniques did not yield sound results (data not shown), likely because permafrost is to a large extent a thermal property and needs a dedicated prediction approach for the occurrence and active layer thickness (Riseborough et al., 2008). Such results could however be combined with predicted SOC values. The predicted depth of the

OL compares well to sampled depths except for Sphagnum and forested wetlands, where it underestimates mean depth and for alpine willow class, where it doubles the pedon mean.

Table 2 near here

The relative landscape SOC storage partitioning is shown in Fig. 7. Birch forest stores 41% of the total SOC covering 41% of the soil area. This is followed by alpine heath tundra that stores 12% of the SOC on 17% of the area. The individual wetland classes have the highest ratio of stored SOC (5.5–6.8%) compared to the area covered (1.2–1.8%) (except *Sphagnum* wetland, where the ratio is 0.7% SOC to 0.1% of the area), while bare ground has the lowest ratio covering 8% of the area and storing only 1.2% of the SOC. This is likely an overestimation.

Figure 7 near here

Eight samples have been radiocarbon-dated to understand longterm C dynamics in the system (Table 3). The oldest sample from the central mire indicates a transition from C enriched mineral sediments to organic peat with an age of 5218 cal yrs

BP. A second phase of increased peat accumulation was dated to 2230–1936 cal yrs BP as found in a profile close to the shores of lake Mellersta Harrsjön and in ombrotrophic waterlogged *Sphagnum* patch in the center of Stordalen mire. For the center of the mire, one sample indicates a marked change in peat composition at $150.84 \pm 0.36$ pMC, which likely corresponds to a change from poor fen to palsa peat accumulation related to permafrost accumulation (Kokfelt et al., 2010).



In the birch forest at mid-slope position two samples from a pedon were dated to modern age and 150 cal yrs BP at the base of the OL on a very shallow soil of ~31 cm depth. In lower located birch forest the base of OL had a modern age and soil at a depth of 20–26 cm was dated to 1345 cal yrs BP.

Table3 near here

## 5. Discussion

Quantitative estimates of SOC, its spatial distribution and longterm dynamics in permafrost environments are a major uncertainty in future predictions of the global C cycle. Therefore, new SOC methods to estimate landscape scale SOC

storage need to be investigated and connected to a temporal framework. This article demonstrates the successful prediction of SOC using machine-learning algorithms at very high spatial resolution ($2 \times 2$m) in a sub-Arctic permafrost peatland environment. Four SOC prediction models were tested. The prediction approach is discussed, followed by insights into environmental controls of SOC, present day SOC stocks and their past and future development.

### 5.1  Predicting SOC using machine-learning algorithms

Of the compared prediction algorithms, the RF algorithm clearly performed best. This applies to all three error criteria: $R^2$, CCC and RMSE (Table 1). The high $R^2$ of 0.939 for the internal validation can be confirmed by external validation ($R^2$ = 0.908), a high value for the CCC and lowest RMSE value. On visual comparison it was the only algorithm that could reflect the expected distribution of SOC following simple thematic mapping and also replicate SOC dominating landforms (peat

plateaus) realistically (Fig. 5). The advantage of tree based machine-learning techniques is that they can cope well with non-linearity and have minimal assumptions about the data (McBratney et al., 2003). This is an important property, as in the northern circumpolar permafrost region only limited environmental datasets of varying quality are available. ANN and SVM are also non-linear methods, however in this case they did not perform equally well as the RF model. The RF model often shows superior performance for regression applications (Li et al., 2011), however, in some environments other models

outperformed RF, for example SVM (Were et al., 2015) or ANN (Taghizadeh-Mehrjardi et al., 2016). Mishra and Riley (2012) showed that Geographically weighted regression (GWR) can be successfully used at regional level to predict SOC stocks in Alaska at 60 m spatial resolution. However, GWR is based on the concept of spatial autocorrelation (Fotheringham et al., 2002). In Abisko, very strong environmental gradients of SOC distribution are found and suggest low spatial autocorrelation. In general, machine-learning algorithms are a very promising approach for regression modeling of

SOC in peatland and permafrost environments with RF providing the best results.

This study highlights two distinctive and related local scale properties of high-latitude permafrost ecosystems, that need to be considered when choosing an appropriate machine-learning predictor. One factor is the strong land cover fragmentation of tundra environments with very small land cover patch sizes (Virtanen and Ek, 2014). In the study area, this is reflected in the occurrence of blockfields, small mires and peat plateaus (Fig. 3). The second factor is a high spatial variability of soil

properties and thus SOC storage, related to the presence or absence of permafrost, peatlands and meter scale periglacial landforms. For example, blockfields are areas with SOC–Tot stocks of <0.1 kg C m$^{-2}$. These are often surrounded by forested areas with around $4 – 8$ kg C m$^{-2}$ or in direct neighbourhood of mire ecosystems with SOC–Tot stocks of ~20–





90 kg (Table 2). Analogous, (Hugelius et al., 2011) showed for a study area in the European Russian Arctic, that a Quickbird based 2.4 m spatial resolution LCC was necessary to locate distinctive peat plateaus that store 30–58% of the ecosystem C dominated by SOC, while covering less than ~20% of the area. In areas with ice-wedge polygons, common in lowland tundra environments, the local scale variability of SOC can be even higher from almost zero on polygon rims to

several tens of kg in polygon centers (Ping et al., 2013, 2015, Siewert et al., 2015, 2016). Thus, any applied machine-learning approach must be able to cope with strong and potentially non-linear environmental gradients, while efforts based on spatial-autocorrelation may fail at local scale.

The results show that prediction using few soil pedons (n = 47 + 10) can provide sound results using machine-learning models. The importance to cover the entire environmental gradient of a study area, including end-members with very low

and very high-values in soil surveys is underlined. The original dataset included only very few data points for the lower range of SOC–Tot values close to <0.1 kg C m$^{-2}$ on blockfields. This was compensated by introducing 0.0 kg C m$^{-2}$ SOC pseudo-training examples. The SOC–Tot may be underestimated in small parts of the mires as the applied RF model cannot predict values beyond the range of the input data, e.g. in case the peat accumulation exceeds the maximum thickness of 138 cm that was sampled. Also, potential shifts in environmental gradients that have not been sampled, i.e. changes in

geology, may have strong influence on SOC storage possibly not covered by this study. These demands on the data are analogous to the ones outlined by Hugelius (2012) for making credible thematic mapping and supports the idea of stratified sampling schemes that explicitly include all landscape types.

Sources of error for the predicted SOC values in this article include limitations of transect sampling as opposed to ideal random sampling. However, given the nature of Arctic environments, this is the most time effective data collection method

providing sufficient amounts of soil pedons for regression analysis, but risks the omission of important environmental gradients not covered by transects. In this article strong dependence of the RF model to the LCC could result from overfitting. Yet, good external validation of the projection indicates reliable results and a single dominating predictor variable has also been reported by other authors (e.g. Hengl et al., 2015). The fast degradation of permafrost in the environment has likely let to temporal differences in the high-resolution predictive datasets. For instance, areas visible as

peat bog in the orthophoto from 2008 have since been submerged by water due to permafrost degradation and may have affected some sampling points. Error propagation from the generation of the LCC and the use of the LCC as input variable and for stratified extraction cannot be out-ruled. Different spatial resolutions of the input data can reduce the spatial accuracy of higher resolution input layers. Future applications of machine-learning methods in this context should investigate spatial optimization of input variables (Behrens et al., 2010; Drăguţ et al., 2009), a forward selection of the input

variables (Ließ et al., 2016) or alternatively and dimension reduction using principal component analysis (PCA) to reduce processing time and collinearity among the environmental variables (Howley et al., 2006). Quantitative uncertainty estimates with confidence intervals for the predicted SOC distribution are an important next step (Hugelius, 2012; Hugelius et al., 2014; Zhu, 2000).

### 5.2 Environmental controls of SOC distribution

A set of 23 environmental variables with varying quality and spatial resolution was used. Land cover is the most important variable to predict the total SOC stock, followed by a group of input variables based on the SPOT 5 image, including NIR/SWIR, NIR and SAVI, and DEM derivatives including TWI and elevation and slope. The importance of NDVI in




combination with TWI to predict SOC has also been reported by Taghizadeh-Mehrjardi et al. (2016). All other variables showed comparatively little relevance. Despite or because of the better resolution, the bands of the orthophoto were not important. The scale of the environmental variable can have significant influence on the prediction accuracy (Behrens et al., 2005; Drăguț et al., 2009). The relevance of land cover for the prediction model likely reflects the amount of information already contained in the LCC and its possibility to reproduce distinct soil bodies and sharp transitions from one land cover to the next. The vegetation sensitive SPOT5 bands and composites complement this with information on vegetation productivity and TWI reflects soil moisture variability and slope potentially catenary position. Micro-site effects such as cryoturbation patterns, wind erosion of the OL on small ridges, or increased accumulation in small pits in the alpine soils (Becher et al., 2013; Klaminder et al., 2009) likely occur at a finer resolution than can be resolved in this study. The input environmental variables were not selected or organized according to specific soil forming factors (McBratney et al., 2003). For example no climatic dataset is included. Yet, Klaminder et al. (2009) find a clear connection of SOC accumulation in dry tundra soils and mean annual precipitation along a transect from Abisko towards the more humid western coast. Therefore, climatic datasets would be necessary for a larger study area. From the visual inspection it seems lower resolution environmental variables reduce the spatial accuracy of the mapping result creating a pixel artifact. An exclusion of all SPOT5 input variables was tested, but reduced model performance (result not shown). This indicates that even lower resolution environmental variables can improve the final prediction if they support higher resolution datasets. Yet, Samuel-Rosa et al. (2015) found that more detailed environmental variables only improved model performance incremental and the cost may outweigh the benefits. Instead, efforts should be placed in more soil pedon data. Most sub-Arctic and Arctic research facilities are located in remote areas and logistics are difficult, thus the main source of uncertainty of many local and regional scale studies on SOC distribution is the low amount of available soil data with only ~10–50 pedons. The latest circumpolar SOC estimate for the top meter is based on only 1778 pedons and reports substantial regional gaps in pedon data, particularly for areas in the High Arctic with thin sediment overburden and for cryoturbated and peatland soils (Hugelius et al., 2014). Similarly, it was shown for the SOC storage in Alaska, that despite 556 existing soil pedons, >300 additional soil pedons are necessary to reflect the entire environmental space (Vitharana et al., 2017). Indeed more soil pedon data from the permafrost regions is urgently needed. Future research should also investigate the use of new predictive variables, such as synthetic aperture radar remote sensing data sensitive to soil moisture, which has recently been used to continuously map SOC at circumpolar scale north of the treeline (Bartsch et al., 2016).

### 5.3 Comparing the present day SOC storage

This study provides the first landscape scale estimate and partitioning of SOC for the Abisko area. A mean landscape storage of $7.9 \pm 8.0$ kg C m$^{-2}$ for the SOC–Tot and $7.0 \pm 6.3$ kg C m$^{-2}$ for the SOC 0–100 is predicted. In the mires of the study area, Klaminder et al. (2008) found organic matter stocks of 30–80 kg m$^{-2}$ on hummocks (peat bogs) and 35–110 kg m$^{-2}$ in hollows (wetlands) which translates into similar amounts of SOC as estimated in this study. Some publications have emphasized higher amounts of SOC in tundra heath ~7–9 kg C m$^{-2}$ compared to birch forest ~4–5 kg C m$^{-2}$ (Hartley et al., 2012; Parker et al., 2015). The sampled and predicted values for these classes are in the same range in this study, but higher amounts of SOC in tundra heath cannot be confirmed. Potentially because the sampling included a wider range of birch forest sites, including areas with a longer time for SOC accumulation. C stocks in lakes





were not included, but sediment cores published by Kokfelt et al. (2010) indicate that lakes in the study area could contain similar amounts of C as the sedge wetland class. Overall, the predicted estimates are in line with previous studies for individual land cover types.

Few studies have estimated mean landscape SOC stocks at local scale in mountainous environments in the permafrost regions. Low storage of SOC was documented by Fuchs et al. (2015) in Tarfala, a sub-Arctic alpine valley 50 km south of Stordalen. The SOC stocks vary from 0.05 kg C m$^{-2}$ to 8.4 kg C m$^{-2}$ for different land cover classes, with a landscape mean SOC storage of 0.9 ± 0.2 kg C m$^{-2}$. Dörfer et al. (2013) estimate the mean landscape SOC stocks for two study areas on the Tibetan plateau to 3.4 and 10.4 kg C m$^{-2}$ for the top 0–30 cm of soil. For the Abisko area a value of 3.9 ± 1.7 kg C m$^{-2}$ is calculated for the same depth interval. A study with a similar environmental setting is presented by Palmtag et al. (2015) who investigated SOC stocks in Zackenberg, NE Greenland. This landscape features a combination of higher barren alpine areas and lower wetland areas including palsas. They found a mean landscape SOC storage of 8.3 ± 1.8 kg C m$^{-2}$, which compares well to this study, despite the location in the High Arctic. This could potentially be explained by decreased decomposition rates and slower C turnover (Hobbie et al., 2000). The estimate for Abisko is considerably lower than the 26.1 kg C m$^{-2}$ for SOC 0–100 in the Northern Circumpolar Soil Carbon Database indicated for this area (Hugelius et al., 2013, 2014). A similar discrepancy has been noted by Fuchs et al. (2015) and Palmtag et al. (2016). The NCSCD is based on thematic mapping using soil polygons for northern Europe with an average area of 205 ± 890 km² and averages many soil pedons for individual soil types across the entire Arctic. Clearly, the large generalization and the thematic mapping approach cannot reflect local scale soil properties in highly diverse permafrost environments necessary to assess local or regional C dynamics. This requires high-resolution mapping approaches using satellite imagery to reflect high landscape variability in permafrost regions (Siewert et al., 2015, 2016). However, predictive mapping of SOC values using machine-learning methods combined with remote sensing data could in the future be used to improve northern circumpolar SOC estimates.

Peatlands store large amounts of SOC while occupying only a small fraction of the total landscape. Northern peatlands are highly relevant to the global C cycle and store large amounts of SOC in boreal forests and tundra regions (Gorham, 1991). Peatlands are restricted to waterlogged conditions. In mountainous terrain they therefore mostly occupy valley bottoms. Subsetting the SOC–Tot map by areas that have a OL–Depth ≥40 cm, corresponding to a common definition of peatlands (Tarnocai and Stolbovoy, 2006), it is found that peatlands represent 3.2% of the total soil area, but 13.9% of the SOC–Tot (Fig. 8). Considering the underestimation of the modeled OL depth for forested wetland areas, these values may even increase. According to the LCC, the area covered by all wetland classes is 6.8 %, but this area stores 25.0% of the SOC. For comparison, the Swedish CORINE land cover dataset (© Lantmäteriet, I2014/00691) that is mainly based on Landsat TM imagery and has a resolution of 25 × 25 m, indicates a wetland area of only 3.3% for the same extent. Extracting the SOC for these areas results in only 11.0% of the landscape SOC–Tot. The difference can be attributed to many small scale wetlands in forest areas not captured in the CORINE dataset. This means an underestimation of wetland areas using a country scale land cover map and SOC stored in wetlands by 3.4% of the total soil area and 14.1% of the SOC respectively.

While the major wetland complexes are mapped at coarse resolution, small ones are clearly omitted and high-resolution approaches are necessary to extract this information. It has to be pointed out that 75% of the landscape SOC is not stored in wetlands, but to a large extent in the birch forest with 40.8% and in alpine tundra heath with 12.1% (Fig. 7). The effect of spatial resolution on the mapping of wetlands and fens has also been pointed out by Hugelius (2012) and Virtanen and Ek



(2014). Yet, the significance of these smaller wetland areas for C cycling at landscape scale has so far found little attention in the literature.

Figure 8 near here

### 5.4 SOC age, past and future development

A major research question is whether Arctic environments have in the past and will be in the future a sink or a source of C. The Abisko area was deglaciated around 9500 cal. yrs BP (Berglund et al., 1996), leaving a glacier forefield like landscape with no SOC. This study finds that peat inception in Stordalen took place around 5200 yrs ago for one site in the central mire, while Sonesson (1972) date it to 6000 yrs ago in the central part of the mire and Kokfelt et al. (2010) to around 4700 in the northern part of the mire. Yet, the $^{14}$C samples from other parts of the mire indicate peat deposition and a transition to

pure peat between 1900 and 2200 cal BP that may have accumulated due to peat erosion followed by continuous peat production. Similarly, Kokfelt et al. (2010) found a change to ombrotrophic conditions and potential permafrost aggradation in the mire taking place around 2800 cal BP and prevailing permafrost conditions between 2650–2100 cal BP. This was followed by a phase of thermokarst and peat erosion and SOC accumulation in surrounding lakes. After 700 cal BP permafrost conditions prevailed and palsa formation took place in the northern part of the mire around 120 cal BP (Kokfelt

et al., 2010). The majority of the sampled SOC in the birch forest has an age of less than 1350 years. No signs of significant SOC burial due to solifluction or cryoturbation processes was found in the transect sampling. Yet, these processes can store significant amounts of SOC in alpine and tundra terrain (Palmtag et al., 2015; Siewert et al., 2016) and are a common phenomenon on some slopes of the area. Becher et al. (2013) found three major periods of burial of SOC in non-sorted circles near Abisko, that coincide with transitions from colder to warmer conditions. These were dated to 0–100 A.D., 900–

1250 A.D. and 1650–1950 A.D. Overall, the bulk of the present day SOC in the study area has accumulated during the past 2000 yrs, both for the peatlands and the birch forest. Despite some episodes of palsa and peat plateau degradation and peat erosion, it can be assumed that the study area has over the time period of the Holocene likely been a C sink, with a significant portion of the SOC stored in labile and temperature sensitive peat plateaus.

At present, permafrost of the peat plateaus of northern Fennoscandia seem to be at a critical thermal limit and close to

collapse. In the Abisko region, permafrost is warming rapidly (Johansson et al., 2011). In Stordalen a decrease of dry peat plateau areas by 10% and an increase of wet graminoid dominated water areas by 17% has been documented for the period from 1970 to 2000 and was likely caused by permafrost degradation (Malmer et al., 2005). Snow manipulation experiments have shown a rapid increase of ground temperatures leading to permafrost degradation and changes in vegetation within only seven years (Johansson et al., 2013). In Tavvavuoma, a peat plateau/thermokarst lake complex located in the sporadic

permafrost zone in northern Sweden, the same trend is observed. Here significant thermokarst lake formation, drainage and infilling with fen vegetation has occurred from 1963 to 2003 (Sannel and Kuhry, 2011). These are significant landscape changes that affect the C balance in peat plateau areas and the permafrost in this area is clearly not in equilibrium with the present day climate. Unless pronounced cooling sets in permafrost degradation in the peat plateau will occur (Sannel et al., 2015).

Many geomorphic systems are not in equilibrium and often exhibit a buffered response to external disturbance such as climatic changes (Bull, 1991). Often geomorphic system response is non-linear and complex (Phillips, 2003). Analogous to paraglacial sediment systems (Ballantyne, 2002; Church and Ryder, 1972), the decay of permafrost will trigger and





condition a set of processes that likely result in a significant geomorphic impact. Past degradation of permafrost in Abisko resulted in redistribution of C rich organic sediments from eroding peat plateaus (Kokfelt et al., 2010). Malmer et al. (2005) suggest that much of the eroded peat from Stordalen has been lost as dissolved organic carbon (DOC). DOC may undergo significant changes in composition while it is in the fen system, as has been shown for Stordalen by Olefeldt and Roulet

(2012). In the alpine heath ecosystem cryoturbation followed periods of climate interruptions causing burial of SOC (Becher et al., 2013, 2015). Lundin et al. (2016) find that the Stordalen catchment is unlikely to be a present day C sink, but rather acts as a source of C. However, Fuchs et al. (2015) argue for the alpine Tarfala valley with very low SOC stocks, that these landscapes will under future climatic changes turn into a sink of C, despite degradation of the permafrost, as biomass will increase and soils develop. In Abisko most of the SOC is stored in birch forest soils. A further shrubification of alpine areas

and a rise of the tree line combined with a positive priming effect could bind more SOC in the above ground vegetation (Hartley et al., 2012). Yet, for most permafrost environments this is unlikely to offset SOC releases from the permafrost and deeper soil layers by mass (Siewert et al., 2015) and may even lead to a release of SOC (Hartley et al., 2012). Also, the role of many minor wetland areas revealed by high resolution mapping has not received sufficient attention. Indeed, a holistic perspective will be necessary to predict how SOC storages will evolve in post-permafrost landscapes in the future.

## 6 Conclusions

Very few studies have applied regression methods or digital soil mapping techniques in sub-Arctic and Arctic permafrost environments to map soil organic carbon (SOC). In general thematic mapping approaches have been favored. The results show promising results using machine-learning techniques to predict SOC in a mountainous sub-Arctic peatland

environment with typical periglacial landforms such as permafrost raised peat plateaus. A random forests prediction model showed the best results in terms of coefficient of determination ($R^2 = 0.908$) and upon visual inspection. This is followed by a support vector machine, while multiple linear regression and artificial neural networks could not sufficiently reflect the fragmented SOC distribution with very strong environmental gradients typical for tundra environments. Digital soil mapping of SOC is a significant improvement over upscaling methods using thematic maps such as land cover

classifications or soil maps. Yet, carefully generated thematic maps remain essential to our understanding of a landscape, its partitioning and patchiness. Thematic maps can be used for stratified extraction of soil properties such as SOC and help to understand these variables at landscape level. This study shows good initial results for the use of machine-learning algorithms to project SOC stocks even with limited ground data typical for remote and less accessible study areas in the Arctic. Future field surveys must pay attention to sample the entire environmental gradient, including low C storage end-

members. This applies to local as well as circumpolar scale, where large regional soil pedon gaps remain especially for low areas low in SOC concentration.

For the Abisko study area, the mean landscape total SOC storage is estimated to $7.9 \pm 8.0$ kg C m$^{-2}$ and the 0–100 cm storage to $7.0 \pm 6.3$ kg C m$^{-2}$. This estimate is significantly lower than the estimates from the Northern Circumpolar Soil Carbon Database, but in line with other high-resolution mapping results of SOC storage in similar environments, indicating

the value of high-resolution upscaling and partitioning studies. Many small wetlands that are not resolved in lower resolution studies are mapped and the results highlight the importance of peatlands and peat plateaus for the total SOC stocks in sub-Arctic environments. The landscape history emphasis that present day SOC stocks represent a snapshot in



time in an ecosystem that is subject to continuous environmental change associated with complex biotic and abiotic ecological adoptions and interactions caused by climate change. Rapid future permafrost degradation in peatlands may lead to erosion and transport of organic sediments into lakes and will likely release these into the carbon cycle. The development of alpine heath and birch forest SOC stocks remains unclear. A holistic approach will be necessary to understand 'post-permafrost' processes and landscape distribution of SOC.

## 7    Acknowledgements

The soil data used in this study was collected as part of the European Union FP7—ENVIRONMENT project PAGE21 (grant number 282700), the NordForsk DEFROST (grant number 23001) project and with the support of the Bolin Centre for climate research (grant to Juri Palmtag). I would like to thank Matthias Fuchs, Jannike Andersson, Juri Palmtag and Robin Wojcik for their assistance during fieldwork. Matthias Fuchs and Robin Wojcik for the analysis of the soil samples. I thank Peter Kuhry for his valuable advices to my work. I greatly acknowledge Gustaf Hugelius for his insightful comments on the manuscript and his continuous support.

## 8    Data availability

The soil data collected for this study, the land cover classification and the maps of predicted SOC will be made available via the Pangaea file repository upon acceptance of the manuscript. The used environmental input datasets are openly accessible to Swedish research institutions.





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




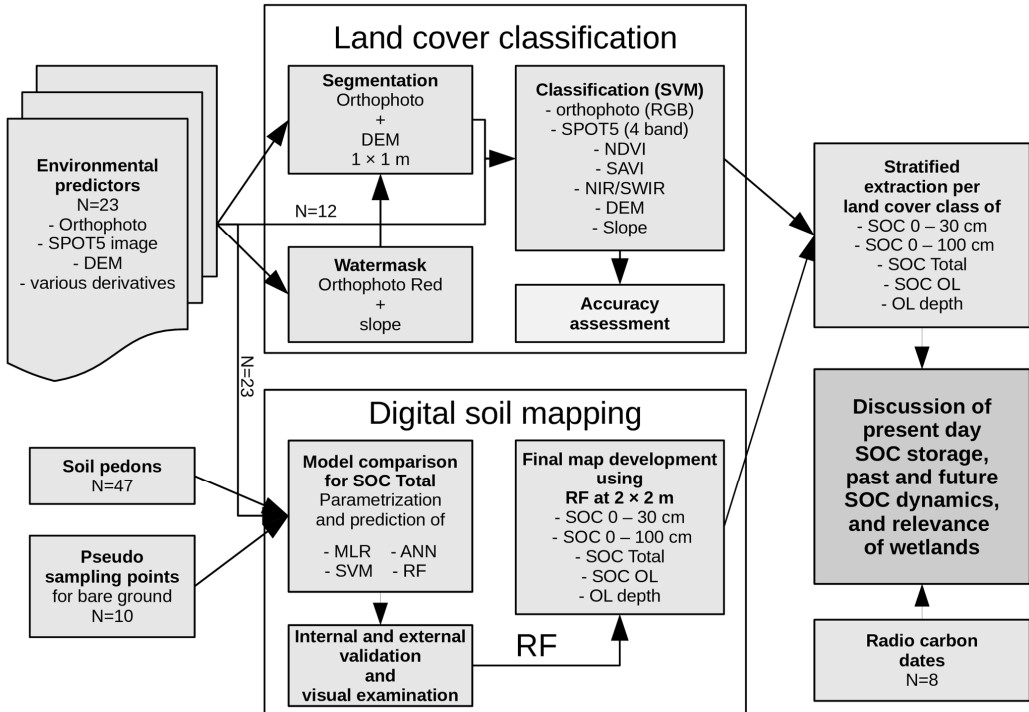

**Figure. 1.** Workflow diagram for this study. Environmental predictor variables are used to generate a land cover classification. Digital soil mapping is performed using soil pedons (supplemented with pseudo sampling points) combined with environmental predictor variables to train prediction models for SOC-Total. The best performing model (RF) is used to develop maps of different SOC depth increments and the OL depth. The results are discussed, with the help of eight radio carbon samples, in the context of present day SOC storage, past and future SOC dynamics and the relevance of wetlands for the SOC storage in permafrost environments.

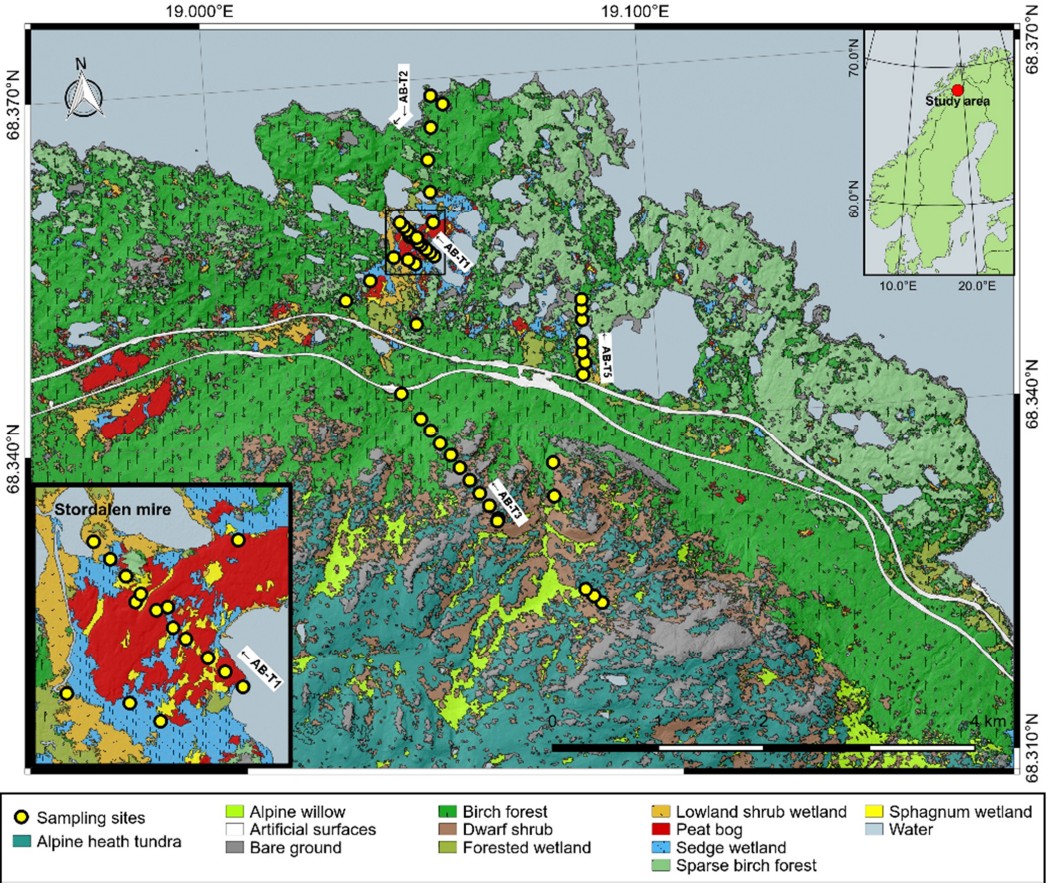

Fig. 2. Top right inset showing the location of the study area. Main view showing the land cover classification for the entire mapping extent. The bottom left inset shows a closeup of Stordalen mire. The beginning and counting direction of the four main sampling transects are marked.




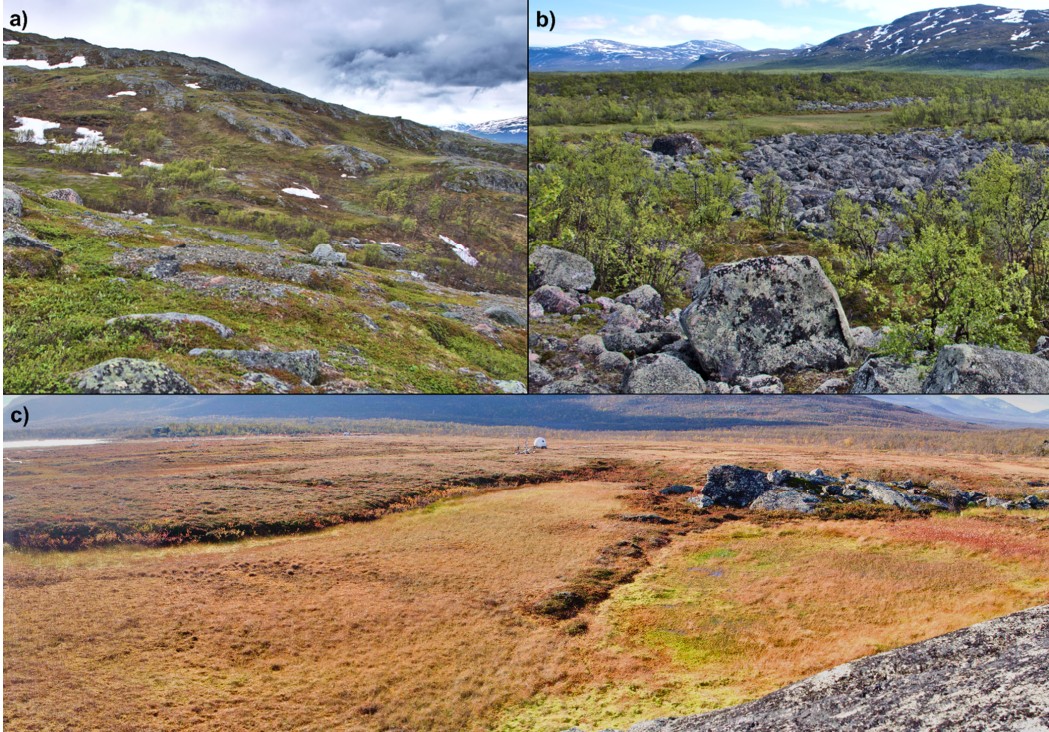

Fig. 3: Photographs exemplifying the study area. a) Alpine landscape mosaic showing several land cover classes including bare ground and alpine tundra heath transitioning into birch forest. b) Lowland landscape mosaic showing sparse birch forest, blockfields and a small wetland in direct neighborhood. c) Raised permafrost peat plateau (left) with sharp transition to Sphagnum dominated wetland areas and exposed bedrock areas (right).






**Fig. 4: A close up of the area near Stordalen mire comparing different maps. a) Illumination corrected orthophoto (© Lantmäteriet, I2014/00691). b) Land cover classification, c) Soil organic carbon storage using a thematic mapping approach. d-g) Maps developed using different machine-learning models. Each map uses all input soil pedons and pseudo sampling points d) multiple linear regression model (MLR), e) artificial neural network (ANN), f) support vector machine (SVM) and g) random forest (RF).**

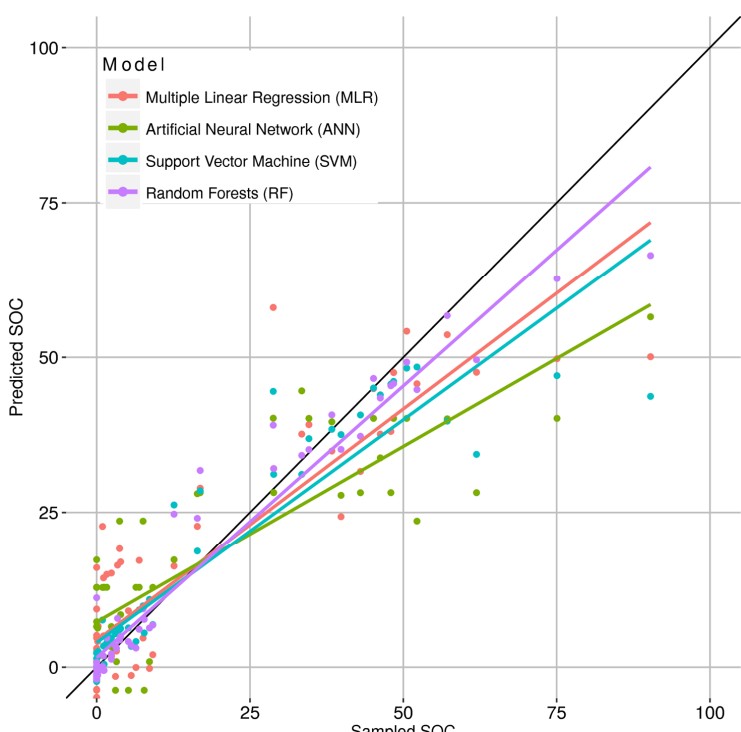

**Fig. 5. Performance of different prediction models comparing observed against predicted total soil organic carbon values (all values in kg C m$^{-2}$) using the full soil pedon dataset including 10 pseudo-training samples for bare ground set to 0.0 kg C m$^{-2}$.**



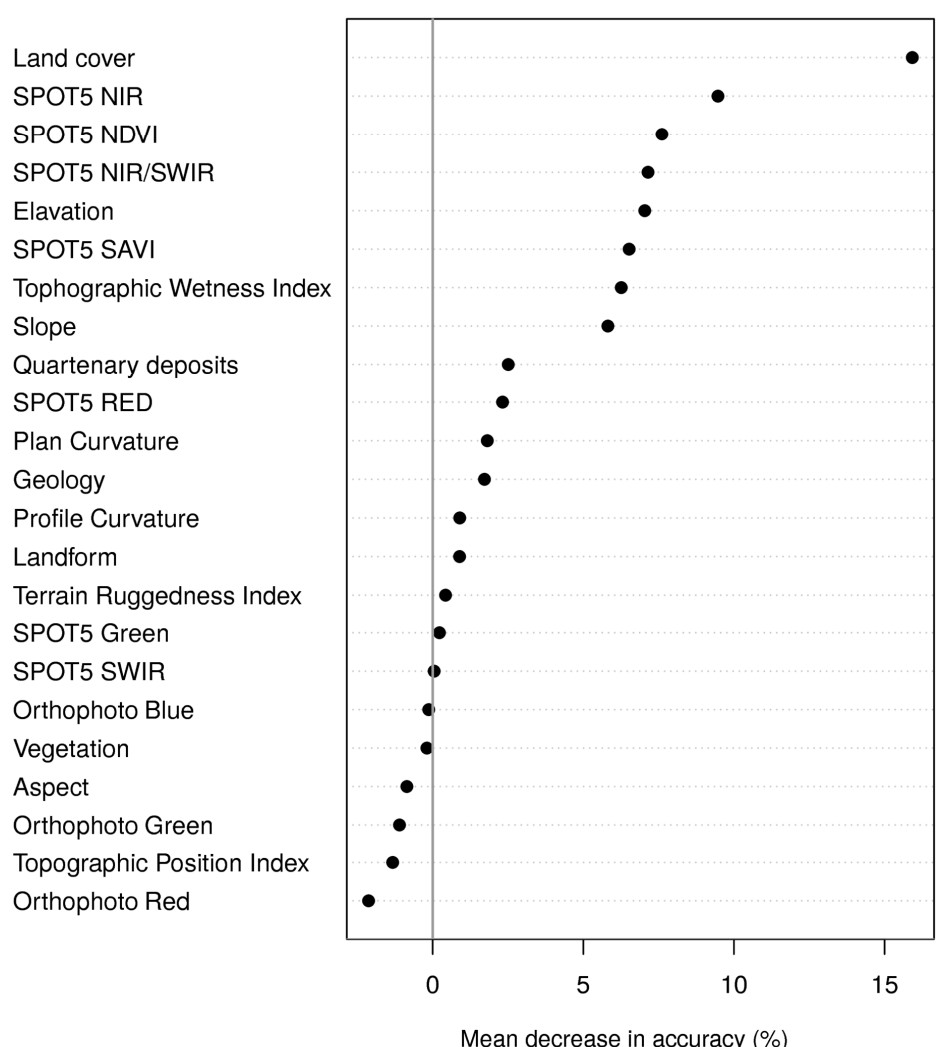

**Fig. 6.** Variable importance for the prediction of total SOC measured as mean decrease in accuracy as a result of permutation of the input variable. The higher the value the more important is the variable.





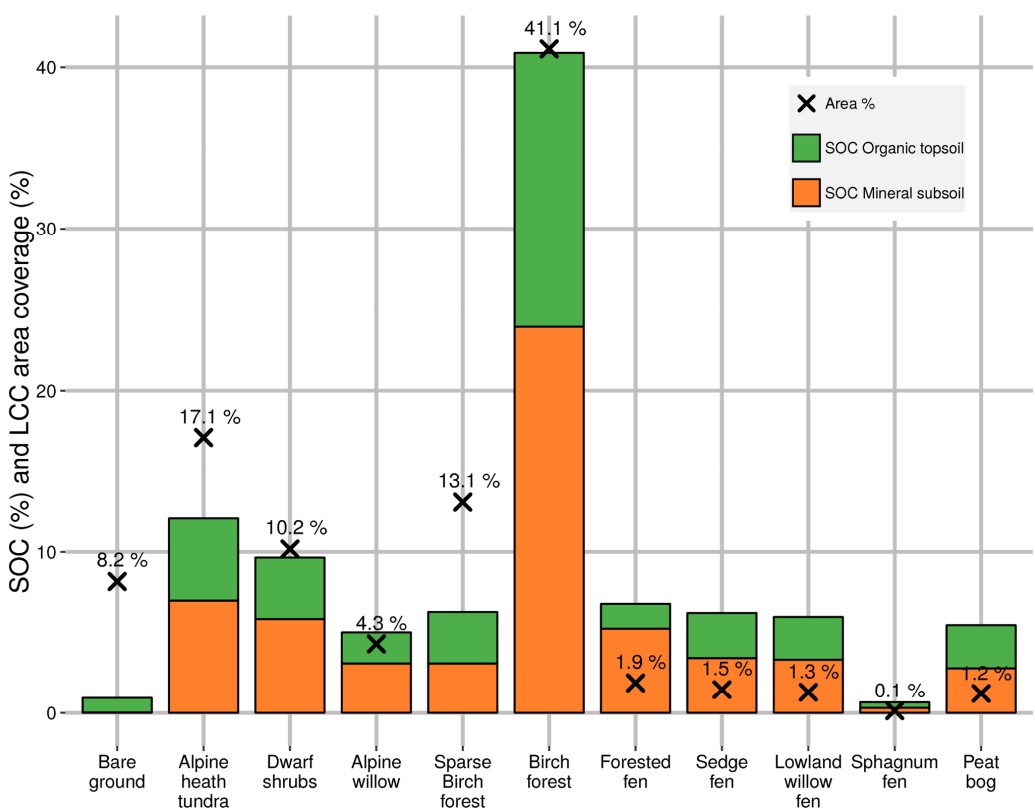

Fig. 7. Partitioning of modeled total soil organic carbon (SOC) storage and respective land cover class coverage in % using a random forest predictor. Height of the column represent the fraction of the SOC–Tot. The mineral SOC is the amount of SOC–OL subtracted from the SOC–Tot. Crosses indicate the percentage areal coverage of the respective land cover class of the total landscape soil area.





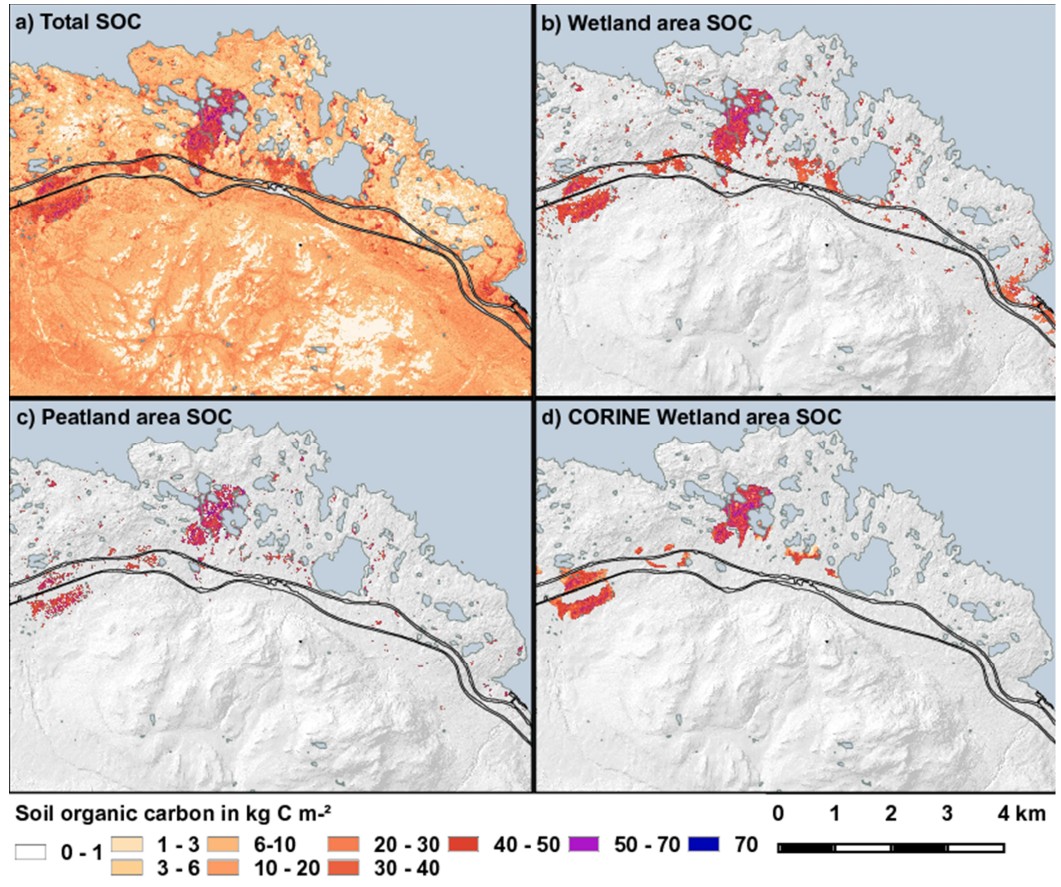

**Fig. 8. Map of the predicted soil organic carbon storage using the random forests model. a) Total soil organic carbon, b) SOC stocks in wetlands classes of the high-resolution land cover classification, c) SOC stocks in peatlands as revealed from modeling the organic layer depth, d) SOC stocks in wetland areas according to the CORINE dataset.**



**Table 1: Validation of SOC-Tot prediction models. The validation is based on the internal (80%) training data subset and on an external (20%) control subset of the SOC input data. The validation is performed with and without pseudo sampling points for bare ground areas.**

| | Including Pseudo sampling points | | | Excluding Pseudo sampling points | | |
|---|---|---|---|---|---|---|
| | $R^2$ | CCC | RMSE | $R^2$ | CCC | RMSE |
| **MLR Int.** | 0.812 | 0.879 | 9.347 | 0.825 | 0.882 | 9.074 |
| **MLR Ext.** | 0.143 | 0.424 | 25.051 | 0.022 | 0.299 | 27.343 |
| **ANN Int.** | 0.843 | 0.872 | 9.1 | 0.823 | 0.858 | 9.560 |
| **ANN Ext.** | 0.352 | 0.547 | 19.739 | 0.229 | 0.440 | 23.362 |
| **SVM Int.** | 0.898 | 0.908 | 7.512 | 0.793 | 0.837 | 10.331 |
| **SVM Ext.** | 0.698 | 0.552 | 17.755 | 0.377 | 0.438 | 21.783 |
| **RF Int.** | 0.939 | 0.925 | 6.482 | 0.949 | 0.945 | 4.986 |
| **RF Ext.** | 0.908 | 0.650 | 15.392 | 0.470 | 0.536 | 19.992 |

$R^2$: coefficient of determination

RMSE: root mean squared error

CCC: Lin's concordance correlation coefficient





**Table 2: Soil pedon properties by land cover class and predicted SOC carbon stocks using a random forest model.**

| Land cover class | Sites (n) | Area in km² | % of Soil Area | Soil pedon data Mean ± StD | | | | | | | | | Random forest predicted values Mean ± StD | | | | |
| --- | --- | --- | --- | --- | --- | --- | --- | --- | --- | --- | --- | --- | --- | --- | --- | --- | --- |
| | | | | Depth Organic layer Mean ± StD | Active layer depth ± StD | SOC Organic layer Mean ± StD | SOC Mineral Mean ± StD | SOC Permafrost Mean ± StD | SOC 0–30 cm Mean ± StD | SOC 0–100 cm Mean ± StD | SOC Total Mean ± StD | SOC Total Min.–Max. | Depth Organic layer Mean ± StD | SOC Organic layer Mean ± StD | SOC 0–30 cm Mean ± StD | SOC 0–100 cm Mean ± StD | SOC Total Mean ± StD |
| Alpine heath tundra | 7 | 8 | 17.1 | 5±3.4 | 37 (1) | 1.4±1.3 | 2.6±2.9 | 0.1±0.3 | 3.5±2.8 | 4±3.1 | 4±3.1 | 1.1–9.2 | 5.5±4.4 | 2.3±1.6 | 4±1.6 | 5.4±3.4 | 5.6±3.8 |
| Alpine willow | 1 | 2 | 4.3 | 5 | – | 1.1 | 6.5 | 0 | 7.5 | 7.6 | 7.6 | 7.6–7.6 | 11±3.7 | 3.3±1.1 | 5.6±0.6 | 8.4±2.6 | 9.2±3.2 |
| Bare ground | 4 | 3.8 | 8.2 | 0.1±0 | – | 0±0 | 0±0 | 0±0 | 0±0 | 0±0 | 0±0 | 0–0 | 1.6±3.3 | 1±1.5 | 1.1±1.4 | 0.9±1.9 | 1.1±2.2 |
| Birch forest | 6 | 19.3 | 41.1 | 9.5±3.9 | – | 2.4±0.4 | 1.8±2.4 | 0±0 | 4±2 | 4.2±2.2 | 4.2±2.2 | 2.4–8.6 | 9.1±4.4 | 3.3±1.6 | 4.9±0.8 | 7.1±3.2 | 7.9±4.3 |
| Dwarf shrubs | 2 | 4.8 | 10.2 | 6±2.8 | – | 1.9±1.3 | 1.4±1.9 | 0±0 | 3.4±3.2 | 3.4±3.2 | 3.4±3.2 | 1.1–5.6 | 8.5±4.3 | 2.8±1.3 | 5±0.9 | 7±3.1 | 7.5±4 |
| Sparse Birch forest | 6 | 6.1 | 13.1 | 5±3.7 | – | 1.7±1.9 | 1.4±1.5 | 0±0 | 3±3.4 | 3.1±3.4 | 3.1±3.4 | 0.2–7.8 | 3.8±3.9 | 2±2 | 3±1.5 | 3.2±2.9 | 3.8±3.6 |
| Forested wetland | 4 | 0.9 | 1.9 | 39.2±34.7 | – | 15.6±22.8 | 24.7±14.1 | 0±0 | 8.9±5.7 | 32.1±15.9 | 40.3±18.7 | 16.4–61.9 | 22.7±8 | 6.8±3.4 | 6.2±0.8 | 24.2±5.2 | 28.6±7.2 |
| Lowland willow wetland | 2 | 0.6 | 1.3 | 46±38.2 | – | 17.5±17.3 | 28.2±18.1 | 0±0 | 9.6±2.5 | 35±6.7 | 45.7±0.8 | 45.1–46.3 | 46.7±13.3 | 15.9±5.2 | 6.9±0.9 | 28.5±5.5 | 36.4±7.8 |
| Peat bog | 9 | 0.6 | 1.2 | 56.9±27.2 | 50±20 (4) | 28.3±20.6 | 23.9±10.4 | 10.1±13.3 | 9.9±4.5 | 40.6±17.9 | 52.2±20.3 | 28.8–90.3 | 42.9±13.2 | 17.6±5.8 | 7.6±1.3 | 28.5±6.9 | 35.3±9.4 |
| Sedge wetland | 4 | 0.7 | 1.5 | 48.8±51.1 | – | 13.3±16.2 | 8±2.4 | 0±0 | 4.6±2.7 | 14.5±8.5 | 21.3±18.2 | 7.7–48 | 43.8±15.3 | 15±5.7 | 5.3±1.6 | 25.9±6.9 | 33.6±9.9 |
| Sphagnum wetland | 2 | 0.1 | 0.1 | 93.5±62.9 | 73 (1) | 22.1±15.4 | 12±14.6 | 8.5±12.1 | 6.6±2.5 | 22.3±6 | 34±0.8 | 33.4–34.6 | 54.3±17.7 | 20.1±6.4 | 7.8±1 | 27.6±6.5 | 38.5±9.3 |
| Anthropogenic | – | 0.8 | 1.6 | | | | | | | | | | | | | | |
| Water | – | 17.4 | 37.1 | | | | | | | | | | | | | | |
| Summary of Study area | 47 | 650.3 | 138.7 | 9.1±1.1 | - | Study area mean ± StD:[a] 2.8±0.3 | 3±0.2 | 0.2±0 | 3.8±0.5 | 5.3±0.5 | 5.8±0.5 | 0–90.3 | Study area mean ± StD: 8.9±9.6 | 3.3±3.5 | 4.3±1.7 | 7±6.3 | 7.9±8 |

[a] Weighted by area, excluding Artificial surfaces and water areas.



**Table 3: Summary of radiocarbon dating.**

| Soil pedon | Depth | Sample description | Lab. no.[a] | Age [14]C | Age cal BP (yrs)[b] |
|---|---|---|---|---|---|
| AB-T1-06 | 14–15 cm | Palsa, base of marked change in peat composition | Poz-59879 | 150.84 ± 0.36 pMC[c] | modern |
| AB-T1-06 | 94–95 cm | Palsa, base of pure peat organics | Poz-59880 | 4565 ± 30 BP | 5218 |
| AB-T1-10 | 72–73 cm | Lowland shrub wetland, base of OL | Poz-59882 | 2215 ± 30 BP | 2230 |
| AB-T2-06 | 48–49 cm | Sphagnum patch, base of OL | Poz-59883 | 1985 ± 30 BP | 1936 |
| AB-T3-07 | 11–12 cm | Alpine birch forest, base of OL | Poz-59884 | 119.16 ± 0.33 pMC | modern |
| AB-T3-07 | 22–28 cm | Alpine birch forest, mineral subsoil | Poz-59885 | 150 ± 30 BP | 150 |
| AB-T3-09 | 18–19 cm | Birch, base of OL | Poz-59886 | 100.82 ± 0.29 pMC | modern |
| AB-T3-09 | 20–26 cm | Birch forest, mineral subsoil | Poz-59887 | 1455 ± 30 BP | 1345 |

[a] Laboratory number of the Radiocarbon Laboratory in Poznan, Poland.
[b] Mean age at 95.4% probability expressed in calendar years before 1950.
[c] Percent Modern Carbon.