# Peer review of "High-resolution digital mapping of soil organic carbon in permafrost terrain using machine-learning: A case study in a sub-Arctic peatland environment"

_Biogeosciences, 2017_

## Referee Comment (RC1) · Anonymous Referee #1 · 21 Sep 2017

This paper discusses a study that developed a high spatial resolution map of soil organic carbon for a sub-Arctic peatland in northern Sweden, using essentially Random Forest algorithms and a suite of environmental variables, including land cover, remotely-sensed vegetation indices, and digital elevation terrain modeling (DEM). The study is relatively straightforward, and demonstrates a reasonable approach for modeling/mapping soil carbon in high northern latitude systems. My only major issue had to do with clarification of the resolutions of the various input datasets, and the ultimate resolution provided by the model/map. That and other minor points are listed here: 1)

So, with regard to the resolution of the inputs and outputs, I found it slightly hard to fol-
low, and I think it might help to put all of the resolutions on Figure 1 (right now only the
orthophoto/DEM and the final map resolutions are on there). If I am understanding this
correctly, the orthophoto is 1m and the DEM is 2m (this is actually slightly misleading
in Figure 1, which has the orthophoto + DEM as 1m – but, I guess that the DEM was
just "down-sampled" to 1m resolution. The SPOT data are either 10m or 20m, and the
minimum size of a land cover classification was $130m^2$, so somewhat consistent with a
SPOT pixel, although it's unclear what the range of extents are for land cover regions.
The final map is then generated at the 2m resolution; why not 1m and utilize the more
resolute orthophoto information? 2) Figure 6 – Does "Mean decrease in accuracy" in-
dicate the accuracy reduction when that variable is removed from the analysis? If so,
make that clear in the figure and caption. 3) Also, it's interesting that the most impor-
tant variable in the analysis was Land Cover, the variable at the coarsest resolution,
followed by three SPOT variables. In fact, you don't get a DEM variable until the 5th-
most important (Elevation), and even then it's unclear that the information is necessary
at the 2m resolution (could be equally useful if aggregated to a coarser resolution). I
know from first-hand experience that these systems can be highly variable in space
over short distances with regard to SOC; however, it's certainly interesting that most of
the variability explained occurs at resolutions of tens of meters, which puts into question
the utility of a 2m resolution map. I think this is worthy of some additional discussion in
the paper – particularly within the context of what is discussed on Page 13, Lines 1-8,
where a fine resolution is necessary to capture the appropriate scale of variability in
SOC. 4) Abstract, Line 10 – add "for SOC quantification" after "evaluated" 5) Abstract,
Line 16 – change "surprising" to "surprisingly" 6) Abstract, Line 19 – add "s" to "scale"
7) Page 2, Line 2 – specify "Northern" high latitudes 8) Page 2, Line 8 – to what depth
is the ∼1300 Pg SOC estimate? 9) Page 2, Line 10 and throughout – be consistent,
either hyphenate "permafrost affected" or not – probably should hyphenate 10) Page
2, Line 24 – remove "a" before "commonly" 11) Page 3, Line 1 – remove the hyphen
from "higher-latitudes" 12) Page 3, Line 15 – I think LCC has not been spelled out yet

in the paper 13) Page 4, Line 19 – How long were the transects (i.e. what was the distance between sampling points)? 14) Page 4, Line 29 – "deeper soil horizons were sampled in 5-10 cm intervals" – what actually were the intervals, and what determined them? 15) Page 5, Line 4 – change "were" to "where" 16) Page 5, Line 6 – should the notation be ">2 mm," if you are referring to the coarse fraction, or are you referring to the soil that is not the coarse fraction? 17) Page 5, Line 13 – add "SOC" before "stored" 18) Page 8, Lines 8-10 – I understand the overestimation of SOC values due to the absence of sample point from bare ground surfaces, however, I just want to clarify the justification for using 0 as the quantity of SOC. First, I'm not sure I know what a "blockfield" is – maybe that's just me, but I think a definition/description would be good. Also, one cause of bare ground in northern high latitudes is cryogenic disturbances (i.e. cryoturbation), and in many cases, these were once vegetated areas that can have quite a bit of SOC. Are these generally uncommon in your study area? In other words, are the dominant bare ground features these blockfields and stone beaches that I imagine have very little SOC? 19) Page 9, Line 4 – add "ed" to "collect" 20) Page 9, Line 10 – remove one "s" from "miss-" 21) Page 11, Line 3 – add "be" after the first "to" and remove the 2nd "to" 22) Page 11, Line 5 and throughout – Sphagnum should be capitalized and Italicized 23) Page 12, Line 26 – don't capitalize "Geographically" 24) Page, 12, Lines 28-29 – I'm not sure that I understand the statement that "very strong environmental gradients" would "suggest low spatial autocorrelation." I would think that strong environmental gradients would lead to high spatial autocorrelation. 25) Page 18, Line 2 – change "adoptions" to "adaptations" – I think that's what you are meaning to say? 26) Page 18, Line 3 – need to reword "release them into the carbon cycle" – even if a carbon pool is stable for a long period of time, it's still in the carbon cycle.

---

## Referee Comment (RC2) · Anonymous Referee #2 · 26 Sep 2017

Siewert presents a study that maps soil organic carbon (SOC) stocks at high spatial resolution (∼2m) for a sub-Arctic study site in Sweden. Four machine-learning algorithms are compared to assess which is best for predicting SOC. The Random Forests method creates the most accurate predictions. The results revealed that vegetation/land-cover type explained most variability in SOC, and thus the spatial distribution of SOC is controlled largely by landcover. On average, landscape scale estimates of SOC are in line with other high-resolution estimates generated at the landscape scale, and these are generally substantially lower than the best available circumpolar estimates generated using thematic maps. Overall the research is good quality and helps advance understanding of spatial variability in high-latitude SOC dynamics. Revisions are required before the manuscript can be considered further for publication.

In general I find the science presented in this study to be sound. However some of the methods could benefit from additional detail. The writing could also be improved to enhance the clarity of the paper. There are quite a few wordy, run-on sentences that are hard to decipher. In other places there are generalities that do no actually convey much information. As a result of these things some very important key points are easy to miss, and this makes the paper seem less important than it actually is. Substantial editing will greatly improve the manuscript. I suspect that it should be possible to reduce the length of the quite a lot without losing any of the current content.

As I mention above, and in specific comments below, aspects of the methods would benefit from additional detail. In particular, the details of several machine learning approaches are unclear. I realize that you use many different data sources, software tools, and analytical approaches, and so there are many details. However, it is becoming more common to publish processing scripts and data (where feasible) with your papers (using a repository such as GitHub, etc. . .). I myself am working to do this, and I encourage others to do the same. This has many benefits, and few downsides.

With regards to the content of the article, one area that I believe should be improved is the discussion of your results in comparison to circumpolar SOC estimates (i.e. NC-SCD). The discrepancy you report is large and seems important, but this is not the first case. Can you discuss potential approaches to bridge these two scales? Would Landsat or MODIS data be appropriate? Since land cover is an important determinant of SOC, it seems as though this could be feasible. Some discussion of how to extend remote sensing methods of SOC prediction to regional and circumpolar scales, and implications for estimates of related SOC stocks would be really useful, especially if the manuscript is edited to improve clarity.

Specific Comments: P2 L14: This seems like an odd place to state the purpose of the articles, especially when it is re-stated in more detail later in the introduction. The introduction should begin with broad context and then gradually narrow to the scope of the present study, whereas this seems to bounce back and forth a bit.

P2 L34-37: Could you elaborate on the evolution of quantitative soils methods, or get rid of this passage. It seems strange to say that methods have changed without at least a brief description of how.

P3 L1-4: Six studies seems like more than a few.

P3 L10-12: Will this really advance knowledge of SOC in all permafrost environments? Perhaps just this particular one, with potential for improved understanding in others.

P3 L14-22: This reads more like methods. It would be better to include this as methods.

P3 L33-34: Probably only need to note the 2002-2011 period just once.

P4 L4-13: This paragraph would fit better with the climatological information, before the detailed soils description.

P5 L4: Typo.

P5 L33-35: This is ambiguous and not necessarily reproducible. Ideally you should publish your scripts/code with the paper.

P7 L11: Did you use the caret package to fit the model as well, or was this just for cross-validation? The methods are a little vague here.

P7 L28: What are 'visual sound results'?

P7 L28-30: This is a run-on sentence.

P8 L2: It would be helpful to specify the number of points (i.e. how many is 20%).

P10 L6-7: This sentence is discussion and doesn't belong in the results.

P10 L8: 'Underestimated opposed' is confusing wording.

P10 L21-27: There is a lot of discussion in here.

P12 L24: In which environments to other algorithms perform better, and why might this be?

P13 L24: Type 'led' not 'let'

P14 L13: How generalizable are these results then?

P14 L17: 'Incrementally'

P15 L15-20: This seems important – can you expand to discuss how these scales might be bridged? Does this mean all areas underestimated? What does this mean for circumpolar SOC stocks?

---

## Referee Comment (RC3) · Anonymous Referee #3 · 3 Oct 2017

General comments:

The study involves the evaluation of different methods for detailed mapping of SOC in permafrost regions. It targets a relevant topic and the methodological approach is sound. The manuscript is well written and thoroughly deals with all sections. Some improvements could be made though. The different machine learning methods were utilized a diverse set of input parameters, including individual parameters (e g spectral bands), derived parameters form single data sources (e g NDVI, TWI) and integrated parameters (landcover/LCC). The single best predictor was LCC which is not surprising

since the LCC integrates several remote sensing sources and also involves manual processing. These diverse types of parameters make it difficult to conclude which raw data sources are most important for SOC mapping. A brief discussion about the importance of different sources could be added to the discussion. Further it would be very interesting to see the performance of LCC alone for mapping as a single predictor. This could be achieved by providing the performance of LCC alone in Table 1. The study focusses on high-resolution mapping (e g 2x2 meters) which is good, but in addition it would be of interest to see how the different methods perform at coarser scales. Unbiased estimate at the 100x100 meter scale or 1x1 km scale is of great importance for global SOC mapping initiatives. A summary of landscape estimates for all the different methods (including LCC) could be added to the results. The SOC distribution in the Abisko area is strongly dependent on the occurrence of peatland areas. In Fig. 4 it can be seen clearly that the modelling mainly separate peatland areas from minerogenic soils. This is not discussed in relation to method performance and implications of the findings.

Detailed comments:

P1 L21: Abisko is misspelled. P2 L15: Describe more specifically which "dramatic changes is peat mires…" that you refer to. P5 L6: I believe it should be ">2mm" instead of "<2 mm". P5 L6: How was the coarse fraction volume determined? P7 L29: Change "visual" to "visually". P9 L24 (also P12 L17): Explain why the external validation was so much superior for RF compared to the other methods. What is the implication of this? P13 L24: Change "let" to "led". P13 L35: Clarify that LCC is an integrated parameter combining many other data sources. P13 L35-: In this section please discuss the inference of your results based on the fact that the distribution of SOC in the Abisko landscape is so strongly dependent on the distribution peatland.

---

## Referee Comment (RC4) · Anonymous Referee #4 · 3 Oct 2017

Author present comparison of four digital soil mapping techniques in predicting high-resolution (2x2m) SOC stocks of sub-Arctic peatland terrain. Study reports that Random forest performed better in comparison to other three techniques used and land cover types derived from a high resolution remote sensing data was the most important predictor of SOC stock variability. Author also report that most of the SOC of study area is relatively new carbon (∼2000 years old).

Author report interesting findings and the outcome should be of interest to a wide readership of Biogeosciences. However, the current manuscript can be improved in multiple

different ways as suggested below:

- The sentence structure at multiple places is awkward so a careful editing is required.
- Its not clear to me how 2x2m spatial resolution for SOC stock was defined? Author seem to have a variety of environmental datasets with spatial resolution ranging from 1 m to 20 m. - I don't agree with the term internal validation used in this manuscript. Using model training dataset as a model validation is not correct. It provides an incorrect metric of map accuracy. For validation, you have to either use the split sample in the beginning (like you did for 20% data) or it has to be take one out approach (cross validation; using remaining samples to predict at the data point by taking out that data point from the model calibration data). - Its not clear to me how land cover data was treated in different models used, were all the land cover types were equally important predictors of SOC? or it was only a subset of all the land cover types? Please provide results. - I will like to see a section on uncertainty in this manuscript. Either calculate the uncertainty or provide a discussion of potential sources of uncertainty involved this study. - The manuscript will benefit if authors can provide reasoning to the observed results. For e.g., why the environmental predictors changed with depths, why certain environmental controllers were significant predictor at certain depth and not other. - How the multicollinearity and non-linear relationships were handled? - Fig. 5 need to be replaced, please remove pseudo sampling points from the plots, provide the number of samples used for model validation. Provide separate plots for 4 mapping techniques using validation samples only. Add R2, RMSE, and CCC values in each plots. - Table 1: Please remove metrices calculated using model calibration datasets, and after adding these values in plots suggested earlier, you will not need this table. In results section, please describe what readers should learn from these map accuracy measures.

---

## Author Comment (AC2) · 8 Nov 2017

Siewert presents a study that maps soil organic carbon (SOC) stocks at high spatial resolution (∼2m) for a sub-Arctic study site in Sweden. Four machine-learning algorithms are compared to assess which is best for predicting SOC. The Random Forests method creates the most accurate predictions. The results revealed that

vegetation/land-cover type explained most variability in SOC, and thus the spatial distribution of SOC is controlled largely by landcover. On average, landscape scale estimates of SOC are in line with other high-resolution estimates generated at the landscape scale, and these are generally substantially lower than the best available circumpolar estimates generated using thematic maps. Overall the research is good quality and helps advance understanding of spatial variability in high-latitude SOC dynamics. Revisions are required before the manuscript can be considered further for publication. In general I find the science presented in this study to be sound. However some of the methods could benefit from additional detail. The writing could also be improved to enhance the clarity of the paper. There are quite a few wordy, run-on sentences that are hard to decipher. In other places there are generalities that do no actually convey much information. As a result of these things some very important key points are easy to miss, and this makes the paper seem less important than it actually is. Substantial editing will greatly improve the manuscript. I suspect that it should be possible to reduce the length of the quite a lot without losing any of the current content. As I mention above, and in specific comments below, aspects of the methods would benefit from additional detail. In particular, the details of several machine learning approaches are unclear. I realize that you use many different data sources, software tools, and analytical approaches, and so there are many details. However, it is becoming more common to publish processing scripts and data (where feasible) with your papers (using a repository such as GitHub, etc. . .). I myself am working to do this, and I encourage others to do the same. This has many benefits, and few downsides. With regards to the content of the article, one area that I believe should be improved is the discussion of your results in comparison to circumpolar SOC estimates (i.e. NCSCD). The discrepancy you report is large and seems important, but this is not the first case. Can you discuss potential approaches to bridge these two scales? Would Landsat or MODIS data be appropriate? Since land cover is an important determinant of SOC, it seems as though this could be feasible. Some discussion of how to extend remote sensing methods of SOC prediction to regional and circumpolar scales, and implications for estimates

of related SOC stocks would be really useful, especially if the manuscript is edited to improve clarity.

**Thank you for this detailed review. The following changes will be made to address the reviewers comments: More detail will be added to the individual methods. However, I don't think that this should mean longer descriptions. There is a lot of literature available on these methods and the interested reader is pointed on several occasions to recent key literature. The writing will be revised throughout the manuscript. The manuscript will also be shortened to emphasis the most relevant results. This will in particular affect the last part of the manuscript that deals with the temporal evolution of the SOC storage. I will consider for future publications to structure my code and workflow in a way that it makes sense to publish the processing scripts.**

**I will add a full discussion section regarding different spatial scales in order to bridge local scale measurements to circumpolar scale. This is in line with the suggestions made by the other reviewers. This would include modeling of the SOC at a scales of 1m, 2m, 10 m, 30 m , 100m and 1000m. Corresponding to available remote sensing data (including Landsat and MODIS) and resolutions used by different model approaches. This will be discussed in context of improvements over the NCSCD at circumpolar level.**

Specific Comments:

P2 L14: This seems like an odd place to state the purpose of the articles, especially when it is re-stated in more detail later in the introduction. The introduction should begin with broad context and then gradually narrow to the scope of the present study, whereas this seems to bounce back and forth a bit.

**The introduction has been restructured and shortened to provide a clearer overview to the topic.**

P2 L34-37: Could you elaborate on the evolution of quantitative soils methods, or get rid of this passage. It seems strange to say that methods have changed without at least a brief description of how.

**The specific passage has been deleted.**

P3 L1-4: Six studies seems like more than a few.

**Thanks. The wording has been changed.**

P3 L10-12: Will this really advance knowledge of SOC in all permafrost environments? Perhaps just this particular one, with potential for improved understanding in others.

**The wording was changed to adopt the perspective of the reviewer:**

**"this will improve our understanding of the SOC distribution and long-term C dynamics in high-latitude ecosystems."**

P3 L14-22: This reads more like methods. It would be better to include this as methods.

**The paragraph has been moved to the methods section.**

P3 L33-34: Probably only need to note the 2002-2011 period just once.

**Changed**

P4 L4-13: This paragraph would fit better with the climatological information, before the detailed soils description.

[Figure]

**The paragraph has been moved.**

P5 L4: Typo.

**Corrected**

P5 L33-35: This is ambiguous and not necessarily reproducible. Ideally you should publish your scripts/code with the paper.

**Thank you for your encouragement. I will consider to publish my scripts in the future.**

P7 L11: Did you use the caret package to fit the model as well, or was this just for cross-validation? The methods are a little vague here.

**Yes, caret was used to fit the model.**

P7 L28: What are 'visual sound results'?

**"Changed to visually meaningful results"**

P7 L28-30: This is a run-on sentence.

**Changed.**

P8 L2: It would be helpful to specify the number of points (i.e. how many is 20%).

**Changed.**

P10 L6-7: This sentence is discussion and doesn't belong in the results.

**The sentence was deleted.**

P10 L8: 'Underestimated opposed' is confusing wording.

**Thank you, the entire paragraph has been edited to improve language.**

P10 L21-27: There is a lot of discussion in here.

**All sentences that discuss the results will be deleted or moved to the discussion section.**

P12 L24: In which environments to other algorithms perform better, and why might this be?

**At this point the general conclusion in the literature is only that no algorithm serves all landscapes. This most likely relates to statistical properties and underlying assumptions of each algorithm and how it can cope with the input data. A sentence was added to underline this.**

**"This indicates that different machine learning algorithms might suit different landscapes and that several algorithms should be compared (Forkuor et al., 2017)."**

P13 L24: Type 'led' not 'let'

**Thanks, Corrected**

P14 L13: How generalizable are these results then?

**Of course there is a limit to what geographical extent a set of input points can be generalized. It is reasonable to assume that a similar environment will feature a similar pattern of SOC distribution, but higher or lower SOC mean values depending on climate.**

P14 L17: 'Incrementally'

**Changed**

P15 L15-20: This seems important – can you expand to discuss how these scales might be bridged? Does this mean all areas underestimated? What does this mean for circumpolar SOC stocks?

**Thank you for your interest. The article will be revised as suggested to include a discussion on scales, how these can be bridged and how circumpolar SOC stock estimates could be improved.**

---

## Author Comment (AC4) · 8 Nov 2017

Author present comparison of four digital soil mapping techniques in predicting high-resolution (2x2m) SOC stocks of sub-Arctic peatland terrain. Study reports that Random forest performed better in comparison to other three techniques used and land cover types derived from a high resolution remote sensing data was the most important predictor of SOC stock variability. Author also report that most of the SOC of study area is relatively new carbon ($\sim$ 2000 years old). Author report interesting findings and the outcome should be of interest to a wide readership of Biogeosciences. However, the current manuscript can be improved in multiple different ways as suggested below:

**Thank you for your review.**

- The sentence structure at multiple places is awkward so a careful editing is required.

**My apologies. The manuscript will be revised throughout with a focus on readability.**

- Its not clear to me how 2x2m spatial resolution for SOC stock was defined? Author seem to have a

variety of environmental datasets with spatial resolution ranging from 1 m to 20 m.

**The spatial resolution of 2x2m was chosen as a compromise between the available input variables, output quality, the benefit of higher resolution and processing time. However, as several reviewers have highlighted interest in the exploration of different resolutions, I suggest to add estimates for 1m, 2m, 10m, 30m, 100m, 1000m. This has been tested and should yield meaningful results. It will however mean a throughout revision of the manuscript.**

- I don't agree with the term internal validation used in this manuscript. Using model training dataset as a model validation is not correct. It provides an incorrect metric of map accuracy. For validation, you have to either use the split sample in the beginning (like you did for 20% data) or it has to be take one out approach (cross validation; using remaining samples to predict at the data point by taking out that data point from the model calibration data).

**The internal validation will be removed from the article.**

- Its not clear to me how land cover data was treated in different models used, were all the land cover types were equally important predictors of SOC? or it was only a subset of all the land cover types? Please provide results.

**The land cover types were treated as equally important predictors. This will be emphasized in the revised discussion.**

- I will like to see a section on uncertainty in this manuscript. Either calculate the uncertainty or provide a discussion of potential sources of uncertainty involved this study.

**A discussion of sources of error is provided on page 13 L 18-33 (original manuscript). The section will update to point out uncertainty and be given a separate heading to make it easier accessible for the reader.**

- The manuscript will benefit if authors can provide reasoning to the observed results. For e.g., why the environmental predictors changed with depths, why certain environmental controllers were significant predictor at certain depth and not other.

**The revised version will contain a more in depth discussion of these topics.**

-How the multicollinearity and non-linear relationships were handled?

**Multicollinearity was tested using a cross-table of the predicting variables. In the revised version, highly correlated predictive variables will be excluded. Non-linear relationships can be handled by the chosen models. See the discussion in section 5.1.**

- Fig. 5 need to be replaced, please remove pseudo sampling points from the plots,

provide the number of samples used for model validation. Provide separate plots for 4 mapping techniques

using validation samples only. Add R2, RMSE, and CCC values in each plots.

**I see the need to replace Figure 5. However, if the figure is replaced according to the suggestion of the reviewer (excluding training and pseudo sampling points) it would mean that it will only be based on 10 validation points per model. This due to the low amount of sample points to start with. Using the full dataset (excluding pseudo sampling points) will provide much more information to the reader than just ten points. The R2 and RMSE can in this case be derived from cross-validation (one out approach) as suggested by the reviewer earlier on.**

- Table 1: Please remove metrices calculated using model calibration datasets, and after adding these values in plots suggested earlier, you will not need this table. In results section, please describe what readers should learn from these map accuracy measures.

**Table 1 has been removed. The information will be added in Fig. 5. Detail was added in the result section to describe what the reader should learn from the measures in terms of accuracy and precision.**

---

## Author Response (AR1)

**Associate Editor Decision: Reconsider after major revisions** (30 Nov 2017) by Susan Natali

Comments to the Author:

Dear Dr. Siewert,

Thank you for your responses to the referees' comments. I agree with the reviewers' comments and your response to address issues related to the spatial resolution of the model. With this and other suggested changes, including editing for readability and inclusion of processing code, I feel that this will be a much improved and publishable manuscript.

Best regard,

Sue Natali

**Response letter to the editor:**

Dear Dr. Natali,

Thank you for your positive feedback to my manuscript. It has now been fully revised. Major changes include:
- A section testing the modelling approach at different spatial resolutions.
- Major revisions of the text addressing all reviewer comments as outlined below and editing for readability.
- link to the processing code at github as suggest by Referee #2

I hope that the manuscript will be judge publishable after these improvements.

I am looking forward to your feedback.

With best regards,

Matthias Siewert

**Response letter to the reviewers:**

Anonymous Referee #1

This paper discusses a study that developed a high spatial resolution map of soil organic carbon for a sub-Arctic peatland in northern Sweden, using essentially Random Forest algorithms and a suite of environmental variables, including land cover,remotely-sensed vegetation indices, and digital elevation terrain modeling (DEM). The study is relatively straightforward, and demonstrates a reasonable approach for modeling/mapping soil carbon in high northern latitude systems. My only major issue had to do with clarification of the resolutions of the various input datasets, and the ultimate resolution provided by the model/map.

That and other minor points are listed here:

1)So, with regard to the resolution of the inputs and outputs, I found it slightly hard to follow, and I think it might help to put all of the resolutions on Figure 1 (right now only the orthophoto/DEM and the final map resolutions are on there). If I am understanding this correctly, the orthophoto is 1m and the DEM is 2m (this is actually slightly misleading in Figure 1, which has the orthophoto + DEM as 1m – but, I guess that the DEM was just "down-sampled" to 1m resolution. The SPOT data are either 10m or 20m, and the minimum size of a land cover classification was 130m 2 , so somewhat consistent with a SPOT pixel, although it's unclear what the range of extents are for land cover regions.
The final map is then generated at the 2m resolution; why not 1m and utilize the more resolute orthophoto information?

**Thank your for this very interesting comment that opens up a different perspective to this work. The resolution of the individual products is now mentioned in Figure 1 (Now Figure 3). A resolution of 2 m for the final model was originally chosen as a compromise between the available input variables, output quality, the benefit of higher resolution and processing time.**

**However, as this point has been mentioned by several reviewers, I ran the model at several spatial resolutions: 1m, 2m, 10m, 30m, 100m, 250 m and 1000 m for the Total SOC and at 1m for different depth intervals. This is in line with the suggestions made by reviewer 3 and 4. The outcome is discussed following the reviewers input with regard to the resolution of different input datasets.**

2) Figure 6 – Does "Mean decrease in accuracy" indicate the accuracy reduction when that variable is removed from the analysis? If so, make that clear in the figure and caption.
**Yes, that is the meaning of this measure. The figure caption has been reformulated to emphasis this:**
*Fig. 1. Variable importance for the prediction of total SOC measured as mean decrease in accuracy of the random forest model if the variable is excluded. The higher the value the more important is the variable.*

3) Also, it's interesting that the most important variable in the analysis was Land Cover, the variable at the coarsest resolution, followed by three SPOT variables. In fact, you don't get a DEM variable until the 5th-most important (Elevation), and even then it's unclear that the information is necessary at the 2m resolution (could be equally useful if aggregated to a coarser resolution). I know from first-hand experience that these systems can be highly variable in space over short distances with regard to SOC; however, it's certainly interesting that most of the variability explained occurs at resolutions of tens of

meters, which puts into question the utility of a 2m resolution map. I think this is worthy of some additional discussion in the paper – particularly within the context of what is discussed on Page 13, Lines 1-8,where a fine resolution is necessary to capture the appropriate scale of variability in SOC.

**I agree with the reviewer in this point. Indeed, lower resolution input variables seem more important than higher resolution input variables. However, I believe that this is much an effect of the validation rather than true value in the spatial prediction using the model. Looking at the resulting maps of SOC in Figure 6, it is clearly visible that information from the DEM has a strong influence on the final map. This seems to be more relevant for fine scale and linear landscape features, while larger homogeneous areas are more influenced by lower resolution input data. The discussion was updated regarding this comment (see section 5.3, 2 paragraph).**

4) Abstract, Line 10 – add "for SOC quantification" after "evaluated"
**Changed**

5) Abstract, Line 16 – change "surprising" to "surprisingly"
**Changed**

6) Abstract, Line 19 – add "s" to "scale"
**Changed**

7) Page 2, Line 2 – specify "Northern" high latitudes
**Changed**
8) Page 2, Line 8 – to what depth is the ∼1300 Pg SOC estimate?
**It is now specified that this includes "soils to a depth of 3 meters and other unconsolidated deposits ". The reader can get more information on this under the specified reference.**

9) Page 2, Line 10 and throughout – be consistent, either hyphenate "permafrost affected" or not – probably should hyphenate
**Hyphenate is now used throughout**

10) Page 2, Line 24 – remove "a" before "commonly"
**Changed**

11) Page 3, Line 1 – remove the hyphen from "higher-latitudes"
**Changed**

12) Page 3, Line 15 – I think LCC has not been spelled out yet in the paper
**LCC has been spelled out on page 1 of the introduction.**

13) Page 4, Line 19 – How long were the transects (i.e. what was the distance between sampling points)?
**The following information was added: "between 50 to 300 m (Fig. 2)". This should enable the reader to understand the sampling layout.**

14) Page 4, Line 29 – "deeper soil horizons were sampled in 5-10 cm intervals" – what actually were the intervals, and what determined  them?
**To be more specific it was added "depending on horizon thickness"**

15) Page 5, Line 4 – change "were" to "where"
**Changed**

16) Page 5, Line 6 – should the notation be ">2 mm," if you are referring to the coarse fraction, or are you referring to the soil that is not the coarse fraction?
**Changed to > 2mm.**

17) Page 5, Line 13 – add "SOC" before "stored"
**Changed**

18) Page 8, Lines 8-10 – I understand the overestimation of SOC values due to the absence of sample point from bare ground surfaces, however, I just want to clarify the justification for using 0 as the quantity of SOC. First, I'm not sure I know what a "blockfield" is – maybe that's just me, but I think a definition/description would be good. Also, one cause of bare ground in northern high latitudes is cryogenic disturbances (i.e. cryoturbation), and in many cases, these were once vegetated areas that can have quite a bit of SOC. Are these generally uncommon in your study area? In other words, are the dominant bare ground features these blockfields and stone beaches that I imagine have very little SOC?
**The following has been added to clarify this:**
**"Originally, all models overestimated SOC contents for bare ground surfaces. These areas include exposed bedrock, blockfields (areas covered by shattered rock fragments with little or no fine substrate; Fig2b) and stone beaches along lake shores (alpine heat tundra with minimal soil development and cryogenic features form a separate class). "**

19) Page 9, Line 4 – add "ed" to "collect"
**Changed**

20) Page 9, Line 10 – remove one "s" from "miss-"
**Changed**

21) Page 11, Line 3 – add "be" after the first "to" and remove the 2nd "to"
**Changed**

22) Page 11, Line 5 and throughout – Sphagnum should be capitalized and Italicized
**Changed**

23) Page 12, Line 26 – don't capitalize "Geographically"
**Changed**

24) Page, 12, Lines 28-29 – I'm not sure that I understand the statement that "very strong environmental gradients" would "suggest low spatial autocorrelation." I would think that strong environmental gradients would lead to high spatial autocorrelation.
**Rephrased to "These sharp transitions in SOC storage between different land covers suggest low spatial autocorrelation at local scale, i.e. little relationship in SOC values between points far apart"**

25) Page 18, Line 2 – change "adoptions" to "adaptations" – I think that's what you are meaning to say?

**Changed**
**Thanks, that's clearly what I meant.**

26) Page 18, Line 3 – need to reword "release them into the carbon cycle" – even if a carbon pool is stable for a long period of time, it's still in the carbon cycle.
**The sentence was reworded to avoid this construct:**
**Rapid future permafrost degradation in peatlands may lead to erosion of organic sediments. This would transfer presently stored carbon into lakes and potentially into the atmosphere.**

Anonymous Referee #2

Siewert presents a study that maps soil organic carbon (SOC) stocks at high spatial resolution (∼2m) for a sub-Arctic study site in Sweden. Four machine-learning algorithms are compared to assess which is best for predicting SOC. The Random Forests method creates the most accurate predictions. The results revealed that vegetation/land-cover type explained most variability in SOC, and thus the spatial distribution of SOC is controlled largely by landcover. On average, landscape scale estimates of SOC are in line with other high-resolution estimates generated at the landscape scale, and these are generally substantially lower than the best available circum-polar estimates generated using thematic maps. Overall the research is good quality and helps advance understanding of spatial variability in high-latitude SOC dynamics. Revisions are required before the manuscript can be considered further for publication. In general I find the science presented in this study to be sound. However some of the methods could benefit from additional detail. The writing could also be improved to enhance the clarity of the paper. There are quite a few wordy, run-on sentences that are hard to decipher. In other places there are generalities that do no actually convey much information. As a result of these things some very important key points are easy to miss, and this makes the paper seem less important than it actually is. Substantial editing will greatly improve the manuscript. I suspect that it should be possible to reduce the length of the quite a lot without losing any of the current content. As I mention above, and in specific comments below, aspects of the methods would benefit from additional detail. In particular, the details of several machine learning approaches are unclear. I realize that you use many different data sources, software tools, and analytical approaches, and so there are many details. However, it is becoming more common to publish processing scripts and data (where feasible) with your papers (using a repository such as GitHub, etc. . .). I myself am working to do this, and I encourage others to do the same. This has many benefits, and few downsides. With regards to the content of the article, one area that I believe should be improved is the discussion of your results in comparison to circumpolar SOC estimates (i.e. NCSCD). The discrepancy you report is large and seems important, but this is not the first case. Can you discuss potential approaches to bridge these two scales? Would Landsat or MODIS data be appropriate? Since land cover is an important determinant of SOC, it seems as though this could be feasible. Some discussion of how to extend remote sensing methods of SOC prediction to regional and circumpolar scales, and implications for estimates of related SOC stocks would be really useful, especially if the manuscript is edited to improve clarity.

**Thank you for this detailed review. The following changes have been made to address the reviewers comments: Some detail was added to the individual methods. However, a lot of literature is available on these methods and the interested reader is pointed on several occasions to recent key literature. The manuscript was edited throughout which hopefully improved the readability. The redundant sections of the manuscript were shortened or deleted. A link to a public github repository was added to the supplement material. The repository contains code relevant to reproducability of the article.**
**An analysis to investigate model performance at different resolutions of 1m, 2m, 10m, 30m, 100m, 250m and 1000m was added. A full section addressing the discrepancy to the NCSCD and the effect of a reduced spatial resolution in the model was added to the discussion section. This includes a discussion on how to bridge both scales and which satellite data could be used to improve circumpolar estimates. This is in line with the suggestions made by the other reviewers.**

Specific Comments:
P2 L14: This seems like an odd place to state the purpose of the articles, especially when it is re-stated in more detail later in the introduction. The introduction should begin with broad context and then gradually narrow to the scope of the present study, whereas this seems to bounce back and forth a bit.

**The introduction has been restructured and shortened to provide a clearer overview to the topic.**

P2 L34-37: Could you elaborate on the evolution of quantitative soils methods, or get rid of this passage. It seems strange to say that methods have changed without at least a brief description of how. **The specific passage has been deleted.**

P3 L1-4: Six studies seems like more than a few. **Thanks. The wording has been changed.**

P3 L10-12: Will this really advance knowledge of SOC in all permafrost environments? Perhaps just this particular one, with potential for improved understanding in others. **The wording was changed to adopt the perspective of the reviewer: "The mapping approach will be discussed with regard to SOC estimation in permafrost regions at local to circumpolar scale."**

P3 L14-22: This reads more like methods. It would be better to include this as methods. **The paragraph has been moved to the methods section.**

P3 L33-34: Probably only need to note the 2002-2011 period just once. **Changed**

P4 L4-13: This paragraph would fit better with the climatological information, before the detailed soils description. **The paragraph has been moved.**

P5 L4: Typo. **Corrected**

P5 L33-35: This is ambiguous and not necessarily reproducible. Ideally you should publish your scripts/code with the paper. **Thank you for your encouragement. A link to a github repository was added in the supplement.**

P7 L11: Did you use the caret package to fit the model as well, or was this just for cross-validation? The methods are a little vague here. **Yes, caret was used to fit the model.**

P7 L28: What are 'visual sound results'? **"Changed to visually meaningful results"**

P7 L28-30: This is a run-on sentence. **Changed.**

P8 L2: It would be helpful to specify the number of points (i.e. how many is 20%). **Changed.**

P10 L6-7: This sentence is discussion and doesn't belong in the results. **The sentence was deleted.**

P10 L8: 'Underestimated opposed' is confusing wording.

**Thank you, the entire paragraph has been edited to improve language.**

P10 L21-27: There is a lot of discussion in here.
**All sentences that discuss the results will be deleted or moved to the discussion section.**

P12 L24: In which environments to other algorithms perform better, and why might this be?
**At this point the general conclusion in the literature is only that no algorithm serves all landscapes. This most likely relates to statistical properties and underlying assumptions of each algorithm and how it can cope with the input data. A sentence was added to underline this.**

**"This indicates that different machine learning algorithms might suit different landscapes and that several algorithms should be compared (Forkuor et al., 2017)."**

P13 L24: Type 'led' not 'let'
**Thanks, Corrected**

P14 L13: How generalizable are these results then?
**Of course there is a limit to what geographical extent a set of input points can be generalized. It is reasonable to assume that a similar environment will feature a similar pattern of SOC distribution, but higher or lower SOC mean values depending on climate.**

P14 L17: 'Incrementally'
**Changed**

P15 L15-20: This seems important – can you expand to discuss how these scales might be bridged? Does this mean all areas underestimated? What does this mean for circumpolar SOC stocks?
**Thank you for your interest. The article was revised as suggested to include a discussion on scales, how these can be bridged and how circumpolar SOC stock estimates could be improved.**

Anonymous Referee #3

General comments:
The study involves the evaluation of different methods for detailed mapping of SOC in permafrost regions. It targets a relevant topic and the methodological approach is sound. The manuscript is well written and thoroughly deals with all sections. Some improvements could be made though. The different machine learning methods were utilized a diverse set of input parameters, including individual parameters (e g spectral bands), derived parameters form single data sources (e g NDVI, TWI) and integrated parameters (landcover/LCC). The single best predictor was LCC which is not surprising since the LCC integrates several remote sensing sources and also involves manual processing. These diverse types of parameters make it difficult to conclude which raw data sources are most important for SOC mapping. A brief discussion about the importance of different sources could be added to the discussion. Further it would be very interesting to see the performance of LCC alone for mapping as a single predictor. This could be achieved by providing the performance of LCC alone in Table 1. The study focusses on high-resolution mapping (e g 2x2 meters) which is good, but in addition it would be of interest to see how the different methods perform at coarser scales. Unbiased estimate at the 100x100 meter scale or 1x1 km scale is of great importance for global SOC mapping initiatives. A summary of landscape estimates for all the different methods (including LCC) could be added to the results. The SOC distribution in the Abisko area is strongly dependent on the occurrence of peatland areas. In Fig. 4 it can be seen clearly that the modelling mainly separate peatland areas from minerogenic soils. This is not discussed in relation to method performance and implications of the findings.

**Thank you for your review. As several reviewers have suggested to investigate modeling at different spatial scales, I added estimates at spatial resolutions of 1m, 2 m, 10 m, 30m, 100m, 250m and 1 km. To keep the article focused this was implemented using the RF model, as it showed overall the best results. I think there is little point to investigate the other models for all scales other than an initial test at 1x1 m. Furthermore, I considered and tested to model the SOC using only the LCC, but the results where not very promising and don't seem to add to the manuscript in a coherent way. A summary of landscape estimates for different resolutions was added to the results in Table one and the predicted maps are shown in the supplement. A brief discussion point was added regarding the differentiation of peatland soils and minerogenic soils in the model. Indeed, very different controls for these two major SOC populations can be imagined. I will keep this notion in mind for future project.**

Detailed comments:
P1 L21: Abisko is misspelled.
**Changed**

P2 L15: Describe more specifically which "dramatic changes is peat mires..." that you refer to.
**Replaced by " Significant changes in surface structure and vegetation in a peat mire..."**

P5 L6: I believe it should be ">2mm" instead of "<2 mm".
**Thanks, changed**

P5 L6: How was the coarse fraction volume determined?
**Added: "determined by sieving of the sample"**

P7 L29: Change "visual" to "visually".

**Changed**

P9 L24 (also P12 L17): Explain why the external validation was so much superior for RF compared to the other methods. What is the implication of this?
**It is hard to explain why exactly one machine-learning method would perform better than others. I don't see any straightforward answer to this question from the literature. RF is generally known to be a versatile algorithm, while other algorithms can perform better in certain situations, but also require very detailed fine tuning. RF seems to be an overall reasonable recommendation.**

P13 L24: Change "let" to "led".
**changed**

P13 L35: Clarify that LCC is an integrated parameter combining many other data sources.
**This is now clarified further down in the paragraph**

P13 L35-: In this section please discuss the inference of your results based on the fact that the distribution of SOC in the Abisko landscape is so strongly dependent on the distribution peatland.

**A section will be added to address this:**
**"In Abisko, the distribution of SOC is defined by the occurrence of peatlands (Fig. 5 and Fig. 8), to the extent that two separate populations of soil pedons can be identified (Table 1). This strong non-linearity may be the reason, why some models perform better. In the future, it should be tested if in such a case separate models for different populations of soil pedons can improve the prediction."**

Anonymous Referee #4

Author present comparison of four digital soil mapping techniques in predicting high-resolution (2x2m) SOC stocks of sub-Arctic peatland terrain. Study reports that Random forest performed better in comparison to other three techniques used and land cover types derived from a high resolution remote sensing data was the most important predictor of SOC stock variability. Author also report that most of the SOC of study area is relatively new carbon (~ 2000 years old). Author report interesting findings and the outcome should be of interest to a wide readership of Biogeosciences. However, the current manuscript can be improved in multiple different ways as suggested below:

**Thank you for your review.**

- The sentence structure at multiple places is awkward so a careful editing is required.
**My apologies. The manuscript was revised throughout with a focus on readability.**

- Its not clear to me how 2x2m spatial resolution for SOC stock was defined? Author seem to have a variety of environmental datasets with spatial resolution ranging from 1 m to 20 m.
**The spatial resolution of 2x2m was chosen as a compromise between the available input variables, output quality, the benefit of higher resolution and processing time. However, as several reviewers have highlighted interest in the exploration of different resolutions, I changed the changed the standard resolution to 1x1 m and added an anlysis on estimates for resolutions at 1m, 2m, 10m, 30m, 100m,250m and 1000m.**

- I don't agree with the term internal validation used in this manuscript. Using model training dataset as a model validation is not correct. It provides an incorrect metric of map accuracy. For validation, you have to either use the split sample in the beginning (like you did for 20% data) or it has to be take one out approach (cross validation; using remaining samples to predict at the data point by taking out that data point from the model calibration data).
**The internal validation was completely removed. The manuscript metrics are now based on cross validation. All values have been updated.**

- Its not clear to me how land cover data was treated in different models used, were all the land cover types were equally important predictors of SOC? or it was only a subset of all the land cover types? Please provide results.
**The land cover types were treated as equally important predictors.**

- I will like to see a section on uncertainty in this manuscript. Either calculate the uncertainty or provide a discussion of potential sources of uncertainty involved this study.
**A discussion of sources of error is provided on page 13 L 18-33 (original manuscript). The section was update to point out uncertainty and was given a separate heading to make it easier accessible for the reader.**

- The manuscript will benefit if authors can provide reasoning to the observed results. For e.g., why the environmental predictors changed with depths, why certain environmental controllers were significant predictor at certain depth and not other.
**The revised version contains insights regarding the influence of different predictors with depth (section 5.3). However, there is of course limited space for such detail analysis.**

-How the multicollinearity and non-linear relationships were handled?
**Multicollinearity was tested using a cross-table of the predicting variables. In the revised version, highly correlated predictive variables are excluded. Non-linear relationships can be handled by the chosen models. See the updated methods and the discussion in section 5.1.**

- Fig. 5 need to be replaced, please remove pseudo sampling points from the plots, provide the number of samples used for model validation. Provide separate plots for 4 mapping techniques
using validation samples only. Add R2, RMSE, and CCC values in each plots.

**I see the need to replace Figure 5. However, if the figure is replaced according to the suggestion of the reviewer (excluding training and pseudo sampling points) it would mean that it will only be based on 10 validation points per model (as the original pedon dataset is rather small). Using the full dataset (excluding pseudo sampling points) will provide much more information to the reader than just ten points. The R2, RMSE and CCC are now derived from cross-validation (one out approach) as suggested by the reviewer earlier on.**

- Table 1: Please remove metrices calculated using model calibration datasets, and after adding these values in plots suggested earlier, you will not need this table. In results section, please describe what readers should learn from these map accuracy measures.

**Table 1 has been removed. The information will be added in Fig. 5. Detail was added in the result section to describe what the reader should learn from the measures in terms of accuracy and precision.**

[revised manuscript text omitted]
 ~~and SAVI. The strong dependence on the LCC likely reflects sensitivity to land cover segmentation. The SPOT5 variables variables are related to the sensitivity of the included bands to vegetation, but also to bare ground cover signature (Andersson, 2016; Huete, 1988; Rouse et al., 1974). Elevation likely reflects the lapse rate of the mean annual air temperature gradient in steep mountainous terrain. The contribution of the TWI data is likely to identify waterlogged conditions along streams and in mires. This is likely supported by slope and is particularly evident for alpine willow communities that are located along flow accumulation pathways.seem toequally littleoverestimation for the lower range of SOC values. an Exclusion of these predictive variables was tested, but generate~~ and five variables were excluded after testing for multicollinearity.

**The variable importance changes for the individual prediction models of SOC 0–30, SOC 0–100, SOC–OL, Depth–OL (Fig. A.1) and SOC–Tot. However, the pattern usually resembles that of SOC–Tot (Fig. 6). Land cover is the most important enviro**nmental variable in all models except for SOC 0–30, where NIR/SWIR is the most important variable followed by NDVI, LCC,  NIR .

Fig. 6 near here

**4.4 SOC stocks and landscape partitioning**

Table 1 shows the landscape partitioning of the sampled SOC pedon values and the predicted SOC values using machine-learning. The landscape mean SOC–Tot storage is predicted to be 98.3 ± 8.0 kg C m$^{-2}$ and 7 ± 6.2 kg C m$^{-2}$ for the top meter ( SOC 0–100) of soil. This compares to 5.8 ± 0.5 kg C m$^{-2}$ for the SOC–Tot  and 5.3 ± 0.5 kg C m$^{-2}$ for the interval 0–100 cm using the LCC for thematic mapping. The highest SOC stock per class is estimated for the *Sphagnum* covered wetland areas (39.5 ± 8.3 kg C m$^{-2}$) followed by the other wetland classes: peat bog (35.5 ± 9.0 kg C m$^{-2}$), lowland shrub wetland (37.2 ± 7.5 kg C m$^{-2}$), sedge wetland

(33.67 ± 9.90 kg C m$^{-2}$) and forested wetland (31.63 ± 67.26 kg C m$^{-2}$). The alpine willow class stores the highest amount of SOC of the remaining non-wetland classes with 10.24 ± 3.24 kg C m$^{-2}$, followed by birch forest (7.98 ± 43.38 kg C m$^{-2}$) and dwarf-shrubs (8.51 ± 43.80 kg C m$^{-2}$). The bare ground class stores the lowest amount of SOC with 1.7 ± 2.23 kg C m$^{-2}$. This represents most likely an overestimation and should be close to <0.1 kg C m$^{-2}$. For SOC–OL, SOC 0–30 and SOC 0–100 similar patterns emerge. Permafrost was encountered in six pedons of which 4 were located in  peat bog with an average depth of 50 ± 20 cm, one in the *sphagnum* wetlands and one alpine tundra heath. The mire permafrost soils were sampled in early September in 2013, while the alpine heath tundra (AL–Depth = 37 cm) pedon was sampled in June and does not represent maximum annual active layer depth. The partition of SOC stored in permafrost is 10.1 ± 13.3 kg C m$^{-2}$ for peat bog soil s and 8.5 ± 12.1 kg C m$^{-2}$ for *Sphagnum* wetland oils. This equals 0.2 ± 0.0 
[revised manuscript text omitted]
.~~potentially catenary position. slope and soil moisture variabilityreflectsTWI and complement this with information on vegetation productivitycomposites The vegetation sensitive SPOT5 bands and . to the next land coverfrom one to reproduce distinct soil bodies and sharp transitions possibilityits contained in the LCC and alreadyThe relevance of land cover for the prediction model likely reflects the amount of information Micro-site effects such as cryoturbation patterns, wind erosion of the OL on small ridges, or increased accumulation in small pits in the alpine soils (Becher et al., 2013;~~

20  ~~Klaminder et al., 2009) likely occur at a finer resolution than can be resolved in this study. The input environmental variables were not selected or organized according to specific soil forming factors (McBratney et al., 2003). For example no climatic dataset is included. Yet, Klaminder et al. (2009) find a clear connection of SOC accumulation in dry tundra soils and mean annual precipitation along a transect from Abisko towards the more humid western coast. While this study can be considered to be representative for a mountainous area with discontinuous permafrost, it will be necessary to includeTherefore, climatic~~

[revised manuscript text omitted]